# Cryo-EM structure of the extracellular domain of murine Thrombopoietin Receptor in complex with Thrombopoietin

Kaiseal T. G. Sarson-Lawrence[1,2,5], Joshua M. Hardy [1,2,3,5], Josephine Iaria[1,2], Dina Stockwell[1,2], Kira Behrens [1,2], Tamanna Saiyed [1,2], Cyrus Tan[1,2], Leila Jebeli [4], Nichollas E. Scott [4], Toby A. Dite[1,2], Nicos A. Nicola[1,2], Andrew P. Leis[1,2,3], Jeffrey J. Babon [1,2] ✉ & Nadia J. Kershaw [1,2] ✉

Thrombopoietin (Tpo) is the primary regulator of megakaryocyte and platelet numbers and is required for haematopoetic stem cell maintenance. Tpo functions by binding its receptor (TpoR, a homodimeric Class I cytokine receptor) and initiating cell proliferation or differentiation. Here we characterise the murine Tpo:TpoR signalling complex biochemically and structurally, using cryo-electron microscopy. Tpo uses opposing surfaces to recruit two copies of receptor, forming a 1:2 complex. Although it binds to the same, membrane-distal site on both receptor chains, it does so with significantly different affinities and its highly glycosylated C-terminal domain is not required. In one receptor chain, a large insertion, unique to TpoR, forms a partially structured loop that contacts cytokine. Tpo binding induces the juxtaposition of the two receptor chains adjacent to the cell membrane. The therapeutic agent romiplostim also targets the cytokine-binding site and the characterisation presented here supports the future development of improved TpoR agonists.

Thrombopoietin (Tpo) is a hematopoietic cytokine that is essential for the maintenance of hematopoietic stem cells and the regulation of megakaryocyte and platelet production[1–4]. As such, its cognate receptor, the Thrombopoietin receptor (TpoR), is expressed on the surface of haematopoietic stem and progenitor cells, megakaryocytes and platelets. Consistent with its key role in haematopoiesis, mutations that affect the Tpo signalling pathway can lead to several blood-related pathologies due to either excessive or impaired TpoR signalling[5,6].

Tpo is predominantly expressed in the liver[7] and is one of ~50 cytokines that signal via the JAK/STAT pathway[8]. This family of cytokines all share the same overall structure, a single 4-helix bundle in an up-up-down-down conformation[9,10]; however, Tpo is unusual in that it also contains an unstructured, C-terminal extension of approximately 180 amino acids that is decorated with both N- and O-linked glycosylation[11]. The role of the C-terminal extension is believed to be independent of the receptor interaction, rather increasing the secretion and/or the circulatory half-life of the cytokine[12,13].

TpoR (originally named c-Mpl protein due its discovery as the cellular equivalent of an oncogene present in murine myeloproliferative leukaemia virus) is a type I, single-pass transmembrane protein, and belongs to the family of class I homodimeric cytokine receptors (for review see ref. 8). All cytokine receptors contain a cytokine-binding homology region (CHR) in their extracellular domain, composed of a pair of fibronectin type-III (FnIII) domains responsible for

[1]Walter and Eliza Hall Institute of Medical Research, 1G Royal Parade, Parkville 3052 Victoria, Australia. [2]Department of Medical Biology, The University of Melbourne, Royal Parade, Parkville 3052 Victoria, Australia. [3]ARC Centre for Cryo-electron Microscopy of Membrane Proteins, Walter and Eliza Hall Institute of Medical Research, 1G Royal Parade, Parkville 3052 Victoria, Australia. [4]Department of Microbiology and Immunology, University of Melbourne at the Peter Doherty Institute for Infection and Immunity, Melbourne 3000 Victoria, Australia. [5]These authors contributed equally: Kaiseal T. G. Sarson-Lawrence, Joshua M. Hardy. ✉e-mail: Babon@wehi.edu.au; Kershaw@wehi.edu.au

recognising the cytokine. TpoR is unusual in that it has tandem CHR domains. The N-terminal, membrane-distal CHR is thought to bind cytokine but may also prevent cytokine-independent activation, as its deletion leads to constitutive activation of the receptor[14]. The function of the membrane-proximal CHR is unknown. TpoR was initially postulated to exist as pre-formed dimers on the cell-surface[15], implying cytokine binding would induce a conformational change in the receptor ectodomain to activate signalling but recent studies have challenged this, measuring <2% dimer formation in the absence of cytokine[16], which suggests that dimerisation by cytokine binding might be enough to trigger signal transduction.

The intracellular domain of TpoR is constitutively associated with a member of the JAK family of kinases[8]. JAKs are catalytically inactive under basal conditions and become auto-activated (in trans) when cytokine binds. Activated JAKs phosphorylate tyrosines in the receptor, which allows binding and subsequent phosphorylation of STATs (Signal Transducers and Activators of Transcription). Phosphorylated STATs then migrate to the nucleus to up-regulate transcription of cytokine-responsive genes[8,17]. JAK2 and TYK2 have been shown to associate with TpoR, with JAK2 playing the dominant role[18], whilst STAT5 is the dominant member of the STAT family involved in TpoR signalling, with additional contributions from STAT1 and STAT3[19]. The MAPK and PI(3)K pathways are also stimulated by Tpo exposure; however, the molecular details and biological impact of this are unclear. The dominant paradigm is that STAT signalling switches on differentiation programmes whilst MAPK and PI(3)K are predominantly proliferative signals. Studies have shown that the orientation of the receptor induced by cytokine binding may be "tunable", allowing one or the other signalling pathway to dominate[20,21].

The presence of too much or too little Tpo signalling leads to a number of disease states. Myeloproliferative neoplasms are a group of diseases caused by mutations that result in constitutive activation of the Tpo signalling pathway[22]. The majority are caused by a single somatic point mutation in JAK2 (JAK2$_{V617F}$[23]) which leads to constitutive activation of the kinase by inducing cytokine-independent dimerisation of TpoR[16]. The second most common type of mutation are frameshift mutations in Calreticulin (CalR)[24,25], which result in it binding to TpoR and being trafficked onto the cell surface where it activates signalling in a Tpo-independent manner[26]. Finally, there are a series of auto-activating mutations in TpoR itself (e.g. W515K, W515L, S505N[27,28]) that lead to enhanced cytokine-independent receptor dimerisation and cytokine-independent signalling[16]. The dependence of signal propagation on JAK2 means that JAK2 inhibitors are used in the treatment of some MPNs[29,30] and can reduce spleen size and improve quality of life. However, their dose is limited by on-target toxicity (anaemia), and they do not reduce the disease allele burden nor effect a cure.

In contrast, deficiencies in TpoR signalling give rise to thrombocytopenias (low platelet number). In a subset of such cases, the use of Tpo agonists can be beneficial. Currently, two classes of agonist are in clinical use: romiplostim is a dimeric, protein-based Tpo-mimetic that targets the cytokine-binding CHR, although its binding mode is not known[31]; small-molecule agents (e.g. eltrombopag, avatrombopag) target the juxta-membrane domains[32].

Despite 30 years since its discovery, Tpo signalling remained until recently one of the few cytokine/receptor systems that lacked structural characterisation. The structure of the human Tpo:TpoR complex was recently solved at 3.4 Å[33]. Here, we present the cryo-EM structure of murine Tpo bound to the ectodomain of TpoR at a nominal resolution of 3.6 Å. The structure shows that Tpo binds to the membrane-distal CHRs to induce receptor dimerisation whilst the geometry of the membrane-proximal CHR is such that the two receptor chains are in contact at the membrane. The cytokine binding site on both receptor chains is located in the hinge region between the two fibronectin domains that comprise CHR1 whilst the analogous region in CHR2

remains solvent exposed. The affinity of receptor for the two sites on the cytokine are vastly different ($K_D$ 100 nM and >10 μM, respectively). The glycosylated C-terminal tail is not required for receptor binding or for signal induction. We confirm that the Tpo-binding site overlaps with the binding site of the biological agonist romiplostim. These data contribute to the structural information required for rational design of a new generation of Tpo agonists and antagonists.

## Results

### Tpo C-terminal domain does not contribute to signalling

Tpo is an unusual cytokine in that it has an unstructured and heavily glycosylated C-terminal extension not seen in other cytokines. Several previous studies have attempted to discern the contribution of the C-terminus of Tpo to signalling, with differing conclusions (e.g.[34,35]). However, to the best of our knowledge, this has not been attempted with highly purified, recombinant proteins generated using the same expression system. Using mammalian cells, we expressed and purified to near homogeneity, full-length murine Tpo, as well the N-terminal 4-helical cytokine domain (Residues 22-184, Tpo$_N$). Both constructs had similar melting temperatures suggesting the glycosylated C-terminus does not have a major effect on intrinsic stability (Supplementary Fig. 1). The full-length protein was heavily glycosylated as demonstrated by SDS-PAGE (apparent molecular weight 70 kDa c.f. theoretical molecular weight in the absence of glycosylation 39 kDa) and mass-photometry (apparent molecular weight 56 ± 16 kDa, Supplementary Fig. 1). In contrast, Tpo$_N$, although too small for mass-photometry, ran close to the predicted molecular weight (20 kDa) by SEC-MALS (22 kDa) and SDS-PAGE (~24 kDa) (Supplementary Fig. 1). The ability of these two forms of Tpo to stimulate differentiation of the M1 murine monocytic leukaemia cell line stably expressing mouse TpoR was tested. Differentiation was assessed by visual examination of colonies after plating in semi-solid agar as described previously[36]. Both full-length Tpo and Tpo$_N$ were similarly potent with an EC$_{50}$ of 10-40 pM (Fig. 1A). Next, the affinity of Tpo and Tpo$_N$ for murine TpoR (monomeric) was measured by Surface Plasmon Resonance (SPR). Tpo and Tpo$_N$ had similar affinities, although the full-length cytokine bound with slightly higher affinity, $K_D$ of 100 ± 20 nM and 160 ± 30 nM respectively (Fig. 1B). These data imply that the majority of the affinity is encoded by the 4-helix bundle domain of the cytokine.

### Tpo:TpoR complex formation

We next attempted to generate the Tpo:TpoR complex in solution and characterise the stoichiometry of the complex using SEC-MALS and mass-photometry. Although reproducible formation of a 1:1 Tpo:TpoR complex was observed at various concentrations and ratios of protein, we were never able to generate the 1:2 stoichiometry (Tpo:TpoR) expected for homodimeric receptors (Fig. 1C, E). This was true regardless of whether Tpo or Tpo$_N$ was used.

We therefore engineered a construct comprising the entire extracellular domain of TpoR with the transmembrane domain replaced with a leucine zipper from *Saccharomyces cerevisiae* GCN4[37], in order to generate a preformed dimer (termed dimeric TpoR). The leucine-zipper was incorporated where the transmembrane domain would be in the wild-type receptor, replacing residues I483 onwards (Supplementary Table 1). This allowed formation of a 1:2 complex as visualised by SEC-MALS (Fig. 1D). Notably, the complex eluted from the SEC column slightly later than apo-dimeric TpoR, implying a reduced hydrodynamic radius and therefore ordering of the dimeric TpoR upon Tpo binding. We interpret this as Tpo engaging with the second copy of the receptor. Mass-photometric experiments (Fig. 1F) indicated that, in the absence of Tpo, the leucine-zippered receptor existed as both monomeric and dimeric species (at an approximate ratio of 1:2) with small amounts of higher order species (<2% trimers and

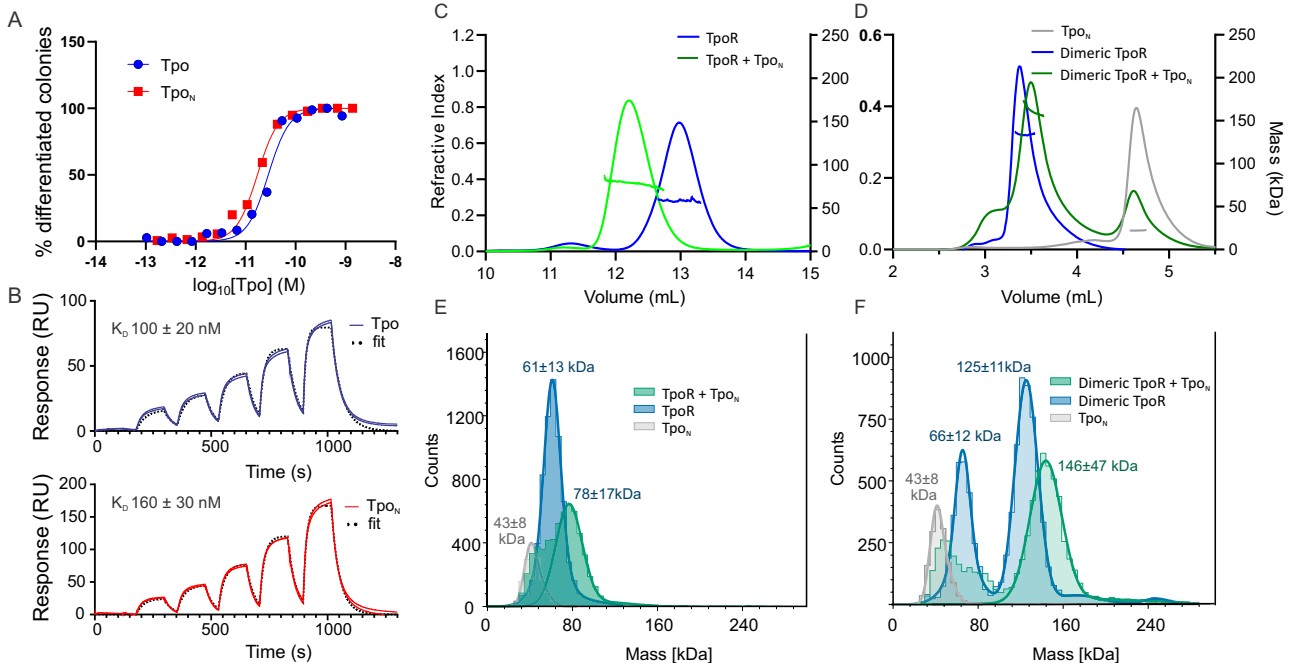

**Fig. 1 | Characterisation of Tpo and TpoN and formation of a Tpo:TpoR complex. A** The cytokine domain of Tpo (TpoN) is equipotent to full-length Tpo. Tpo and TpoN both stimulated differentiation of M1(TpoR) cells with an EC50 of 10–40 pM. The results are representative of two independent experiments using separate batches of cytokine. **B** Sensorgrams showing binding of TpoR ectodomain to immobilised Tpo (full length or cytokine domain) using SPR. Both cytokine species bound with similar affinity and off-rates. Results shown are two technical replicates representative of two independent experiments. A two-fold dilution series of receptor was injected over the chip surface, the highest concentration was 250 nM. **C, D** SEC-MALS analysis of TpoR (monomeric and dimeric) in the presence and absence of TpoN. **E, F** Mass photometry of 100 nM receptor (monomeric or dimeric) in the presence and absence of a 2-fold molar excess of TpoN. TpoN is of insufficient mass to be detected, but there was a small amount of signal visible for a presumably dimeric species. In the presence of TpoN, both monomeric and dimeric receptors yielded an observed particle mass consistent with a single molecule of cytokine being bound. Theoretical molecular weights (in the absence of glycosylation): TpoR (52 kDa), TpoN (20 kDa), dimeric TpoR (118 kDa).

tetramers) also detectable. The addition of a twofold molar excess of Tpo to the leucine-zippered receptor led to the formation of the expected 1:2 complex (cytokine:receptor monomer) (Fig. 1F).

## Cryo-EM reconstruction of the Tpo:TpoR signalling complex

To provide insight into the assembly of the Tpo:TpoR complex, we sought to determine the structure of the ectodomain of the receptor in complex with Tpo. To stabilise the 1:2 complex, we again used the Leucine zippered ectodomain (TpoR1-482-LeuZ). As there was no significant effect of the C-terminal domain of Tpo in any of our assays, we used only the 4-helix bundle (TpoN) for structural studies.

TpoN and TpoR were combined, passed down a size exclusion column and the fractions of highest-purity containing the TpoN:TpoR complex, as analysed by SDS-PAGE, were pooled. The stoichiometry of the complex was validated by SEC-MALS. Analysis of the complex by single particle cryo-electron microscopy yielded a 3D reconstruction with a global resolution of ~3.6 Å (Table 1, Fig. 2A, Supplementary Figs. 2, 3). The resolution was affected by preferential orientation of the complex, which was partially overcome through the addition of the cationic detergent cetyl-trimethylammonium bromide (CTAB) before grid-preparation. An initial model was derived through combining an AlphaFold2 model of the receptor and the crystal structure of human Tpo (PDB ID: 1V7N)[38] as the starting point for refinement and subsequent modelling of the complex. Analysis of the resultant structure revealed a 1:2 Tpo:TpoR complex, consistent with other members of the homodimeric class I cytokine receptors (Fig. 2).

## The structure of the Tpo:TpoR signalling complex

The N-terminal domain of murine Tpo is essentially identical to the crystal structure of human Tpo (PDB ID: 1V7N[38] chain I, RMSD 1.7 Å over

145 residues, DALI[39]), forming the typical cytokine fold of a 4-helix bundle in an up-up-down-down configuration (Fig. 3G). There are two disulphide bonds, one links helices A-D (C28-C172) and the other (C50-C106) links helices A-C. Differences between the mouse and human structures are seen in the AB loop, which contacts the receptor, and the CD loop, which contacts the complexed Fab in the structure of human Tpo (Fig. 3G). Additional density is observed for 5 N-terminal residues of murine Tpo, and the orientation of residues near the C28-C172 disulphide is also altered due to interactions with the receptor.

TpoR is composed of four fibronectin(III) domains that form two typical CHR modules (Fig. 3A)[8]. The complex is ~120 Å in length on its longest axis (perpendicular to the membrane), nearly twice that of Epo:EpoR (~70 Å). The membrane-distal CHR formed by D1 and D2 contains the cytokine-binding site that is formed by loops at the hinge region between the Fn(III) domains (Fig. 3C). This site on each chain interacts with opposite sides of the cytokine (site I and site II) (Fig. 4A). There is a 33 residue TpoR-specific insertion (197-229) in D2 between β-strands D and E of the Fn(III) domain, which is present in many species but with minimal sequence homology and considerable variations in length (Fig. 3A, B, E). The loop is predicted to be unstructured and consistent with this, most of this loop (200–227) is not visible in our structure in chain A (chain for Site I). However, in chain B, (site II chain) there was weak, near-continuous density that could be attributed to this insertion, and it was possible to model the entire loop into the density, though there is significant ambiguity due to the poor quality of the map in this region (Supplementary Fig. 4). Notably, the loop appears to contact helix A of the cytokine and may contribute to site II (we have termed this site IIb, Fig. 4A). Based on the observable density, we hypothesise that three disulphides arise from this loop and adjacent regions (C193-C233, C194-C315, C211-C314) (Fig. 3A, D). Two of these disulphides link D2 to D3. Due to the relatively poor quality of the

**Table 1 | Cryo-EM data collection, refinement and validation statistics**

| | Tpo:TpoR (EMDB-41805) (PDB 8U18) |
|---|---|
| Data collection and processing | |
| Magnification | 105,000 |
| Voltage (kV) | 300 |
| Electron exposure (e⁻/Å²) | 50.57 |
| Defocus range (μm) | 0.4–3.6 |
| Pixel size (Å) | 0.833 |
| Symmetry imposed | C1 |
| Initial particle images (no.) | 2,353,523 |
| Final particle images (no.) | 745,000 |
| Map resolution (Å) | 3.6 |
| FSC threshold | (0.143) |
| Map resolution range (Å) | 3.6–5.0 |
| Refinement | |
| Initial model used (PDB code) | AlphaFold2 model |
| Model resolution (Å) | 3.6 |
| FSC threshold | (0.143) |
| Map sharpening $B$ factor (Å²) | N/A (Local sharpening) |
| Model composition | |
| Non-hydrogen atoms | 7795 |
| Protein residues | 1024 |
| Ligands | BMA: 2, NAG: 8, MAN: 2 |
| $B$ factors (Å²) | |
| Protein (min/max/mean) | 57.79/175.11/99.82 |
| Ligand (min/max/mean) | 101.84/143.80/127.18 |
| R.m.s. deviations | |
| Bond lengths (Å) | 0.003 |
| Bond angles (°) | 0.652 |
| Validation | |
| MolProbity score | 1.18 |
| Clashscore | 3.73 |
| Poor rotamers (%) | 0.77 |
| Ramachandran plot | |
| Favoured (%) | 97.9 |
| Allowed (%) | 2.1 |
| Disallowed (%) | 0.0 |

map in this region, we attempted to validate the disulphide linkages by mass spectrometry (enzyme digest/LC-MS-MS). It is not possible to directly observe the disulphide linkages proposed due to there being two pairs of adjacent cysteines linking three peptides and interpreting the MS-MS spectra of such complex fragments is, to the best of our knowledge, not currently possible. However, this assignment is supported by an evolutionary comparison of the residues involved. Marsupial TpoR homologues lack a single pair of cysteines (equivalent to C211, C314), suggesting that these may form a disulphide bond in other species that have them (Fig. 3B).

The D2-D3 interface (Fig. 3D), which connects the two CHRs, is compact, with the AB loop of D2 nestling on top of the F and G stands of D3, bordered by the CD loop of D3 which is pinned to D2 via a disulphide between C314 and C193 (Fig. 3E). The orientation of these two Fn(III) domains with respect to each other is unusual and is only shared by IL3RA (D1-D2) and IL5RA (D1-D2) according to DALI[39] although the specific interactions responsible are not conserved across these receptors (Supplementary Fig. 5).

D3 and D4 form the membrane proximal CHR2. CHR2, and in particular D4, show the poorest resolution within the structure

(Supplementary Fig. 3A) and displayed a degree of flexibility (Supplementary Movie 1). We were able to enhance the map in this region using 3DFlex[40], but sidechain density for residues in D4 remain largely absent. Despite the poor resolution, it appears that D4 contributes to homodimerization through a D4:D4 interface with an estimated buried surface area of approximately 110 Å² per chain (PISA[41]) (Supplementary Fig. 6). The two TpoR chains in the complex overlay with a RMSD of 1.6 Å over all residues (DALI) despite asymmetrical interactions with the cytokine, which we interpret as implying that TpoR exists as a rigid unit which does not substantially change conformation (but may of course change orientation) upon cytokine binding. CHR1 and CHR2 are very similar in terms of structure and overlay with a backbone RMSD of 4.9 Å over 139 residues (Fig. 3F). Whilst the hinge region in CHR1 binds cytokine, the same region in CHR2 is solvent exposed. There was clear density for three N-linked glycosylation modifications in both chains at N117, N178, and N349, and additional density at Cδ of W465 consistent with mannosylation at this site (Supplementary Fig. 7). Glycosylation state was validated by mass spectrometry (Supplementary Data 1), which confirmed there are three modified asparagine residues, with low oligomannose content, indicating that the glycans are in the mature glycoforms expected from Expi293 cells. Tryptophan mannosylation was detected on several tryptophans, including $W_{465}$ and $W_{468}$ in the WSAWS motif (Supplementary Fig. 8), consistent with observations of the human protein[33,42].

### Cytokine–receptor interfaces

**Site I.** Both site I and site II cytokine receptor interfaces were very well resolved (Fig. 5, Supplementary Fig. 3A). The high-affinity Site I interface is centred around a cluster of hydrophobic residues presented by both receptor and cytokine (Fig. 5A). The binding site is formed from the AB-loop and helix D of the cytokine and the D1-D2 hinge of the receptor. Several residues contribute to complementary hydrophobic surfaces including: F67, L69 (AB-loop), G158, F162, L165, and V166 (Helix D) of Tpo; and L103, F104, I161, F164 and L257 of TpoR. F45 is a key part of this hydrophobic network on the receptor, albeit without directly contacting cytokine. Notably, F45, L103 and F104 are almost invariant across species, F164 is highly conserved, I161 and L257 are more variable but always hydrophobic, consistent with their position on the edge of the hydrophobic cluster, with more room to accommodate alternate residues (Supplementary Fig. 9).

A number of hydrogen bonds and salt bridges extend the interface of site I, which has a total buried surface area of -870 Å² (PISA[41]). Although the resolution precludes a complete definition of the entire polar network, sidechain density for Tpo is particularly clear (Fig. 5A). R161 and R157 form hydrogen bonds with D253 and D128/V255/S256 respectively. S68 is poised to make H-bonds with backbone carbonyls of residues E100 and V101 of receptor, although sidechain density for this residue is weak. On the receptor, R102 is well resolved and forms a salt bridge with D66 of Tpo. Again, the majority of these residues are highly conserved on both cytokine and receptor (Supplementary Fig. 9). Additional hydrophilic interactions are also possible based on proximity, but we have not commented on those where sidechain density is absent. This combination of hydrophobic core surrounded by numerous additional hydrophilic interactions is also seen for Epo:EpoR[43]. Indeed, there is significant similarity between the Epo and Tpo site I binding sites. This differs from prolactin[44] and growth hormone[45], which have a smaller hydrophobic core of two tryptophans and a much more extensive network of hydrogen bonds.

**Site II.** A similar hydrophobic interaction extended by numerous polar contacts is observed for the weaker Site II interaction site (Fig. 5B). Although the total buried surface area is similar for Site II (-870 Å² (PISA[41]), the number of residues involved in hydrophobic interactions is reduced and there are fewer hydrogen bonds. This binding site on the receptor overlaps significantly with site I. The interaction is again

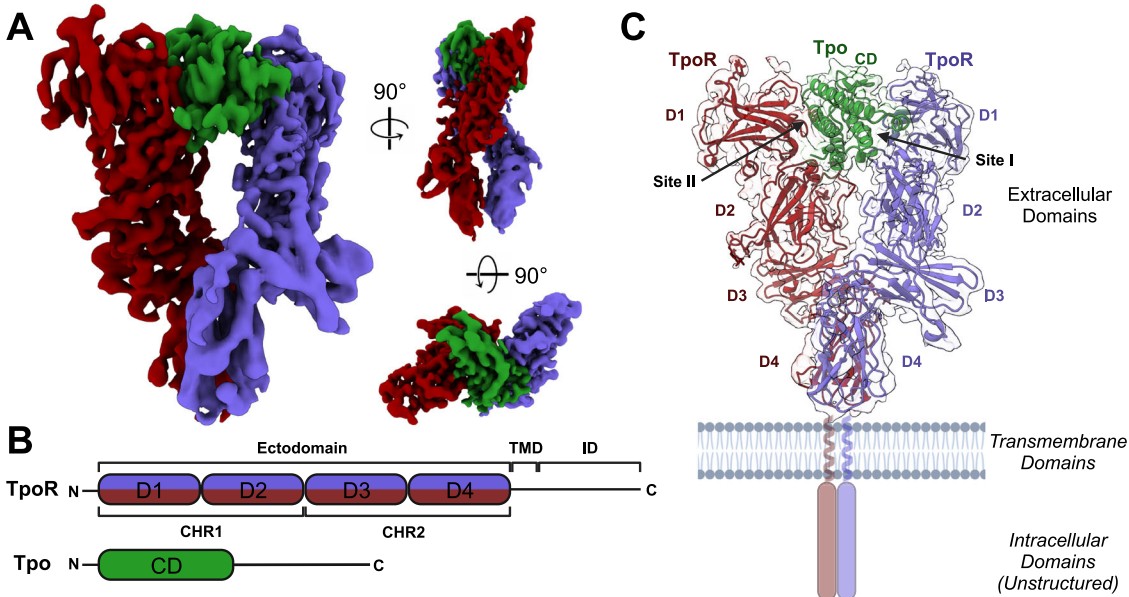

**Fig. 2 | Structure and organisation of the Tpo:TpoR signalling complex. A** Cryo-Electron Microscopy reconstruction of the extracellular domains of the Tpo:TpoR complex resolved to 3.6 Å. Cytokine is coloured green and the two chains of TpoR are coloured purple and red. Two perpendicular views are shown as indicated. **B** Domain organisation of TpoR and Tpo coloured as in (**A**) and (**C**). TMD = transmembrane domain, ID = intracellular domain, CHR = Cytokine receptor Homology Region, CD cytokine domain. **C** Tpo:TpoR complex in a cartoon representation with glycans shown as sticks. The transmembrane and intracellular domains are illustrated here (not to scale) but in the cryo-EM sample they were replaced by a leucine zipper, which was unresolved in the structure. Created using BioRender.com.

centred around residues L103 and F104, F164, and L257 on TpoR, but I161 is not involved. On the cytokine, the interface is on the opposing side to site I, involving helix A and helix C: P26, L32, L120, A124 appear to be the key residues. F105 (TpoR) and L122 (Tpo) may also contribute.

The most clearly resolved hydrophilic sidechains on Tpo are R31 (likely interacts with TpoR-D253), R99 (interacts with TpoR-D99) and R119, which does not form a salt bridge, but rather stacks against the invariant R102 on TpoR (Fig. 5B). D29 is also poised to contribute to the hydrophilic network, but the sidechain orientation and, therefore the H-bond network it forms, is not clear.

There is additional density adjacent to site II that comes from the insertion in D2 of TpoR, which may extend the site II interaction interface with additional regions of helix A (Supplementary Fig. 4). We have termed this site IIb. The tip of the D2 loop contacts Tpo H41 however, due to poor resolution the precise identification of the TpoR residues involved is ambiguous.

**Biochemical analyses of Site I and Site II.** As described in Fig. 1B, the affinity of Tpo and $Tpo_N$ for TpoR was 100 nM and 160 nM, respectively as determined using SPR. However, as it was unclear whether this represented site I, site II or a combination of both, further experiments were performed. When dimeric TpoR was passed over immobilised Tpo it bound with much higher apparent affinity ($K_D$ 1 nM) and with a demonstrably slower off-rate than when using monomeric TpoR (Fig. 6A, B). We interpret this as the dimeric receptor engaging both site I and site II of Tpo on the chip surface (Fig. 6E). Interestingly, this high affinity interaction was also observed when the reciprocal experiment was performed (soluble Tpo and immobilised TpoR) no matter whether monomeric or dimeric TpoR was immobilised (Fig. 6C, D). Presumably, the proximity of individual TpoR chains on the chip surface allows the Tpo analyte to interact with two chains simultaneously (Fig. 6E). This was true even at the lowest surface density that still yielded observable signal. We infer that the $K_D$ of site I is ~100 nM whilst the $K_D$ of site I/II combined is ~1 nM. This infers a very low affinity for site II and is consistent with our inability to observe a 1:2 Tpo:TpoR

complex by SEC-MALS and mass-photometry unless the receptor ectodomain is first dimerised.

Although dimerising the receptor allowed us to measure a 100-fold higher binding affinity, there is still a large discrepancy between the $EC_{50}$ of 40 pM observed in our cell-based assays (Fig. 1A), and the 1 nM affinity observed by SPR (Fig. 6B). Although we inserted the leucine zipper at the sequence position where the transmembrane domain would be, it is possible that the orientation induced by the leucine zipper used to dimerise the receptor extracellular domain was restraining the receptor and reducing the overall affinity of the cytokine:receptor complex. We therefore generated four new leucine zippered constructs, each with one additional amino acid at the N-terminus of the zipper, to allow for one full helical turn and therefore multiple orientation of the receptor ECD. When tested by SPR, one of these forms (+2 residues) showed compromised binding to Tpo however none of them bound Tpo with higher affinity than the original construct (Supplementary Figs. 10, 11).

Based on the structure, mutants of Tpo were generated that were predicted to have compromised site I or II binding. The site I mutant F162E, which inserts a charged residue at a hydrophobic interface (Supplementary Fig. 12) completely abrogated binding to monomeric or dimeric TpoR using the standard concentrations (Fig. 7A-B), suggesting that site I was disrupted and any residual affinity due to site II alone was too low to be measured. We then examined whether we could use high concentrations of the F162E mutant to determine the affinity of site II, however even concentrations as high as 10 μM did not lead to quantifiable data. This suggests that the affinity of the site II interaction is >10 μM. To disrupt site II, we mutated D29 to glutamate and tyrosine (Supplementary Fig. 12). In contrast to F162E, both site II mutants (D29E, D29Y) retained TpoR binding but showed a compromised affinity (5-10-fold weaker) for the dimeric receptor, with an increased off-rate (Fig. 7C). Although the affinity of the D29 mutants was only 5-10-fold decreased relative to the WT, the ability of these mutants to induce differentiation of M1(TpoR) cells was decreased 100-fold (Fig. 7D).

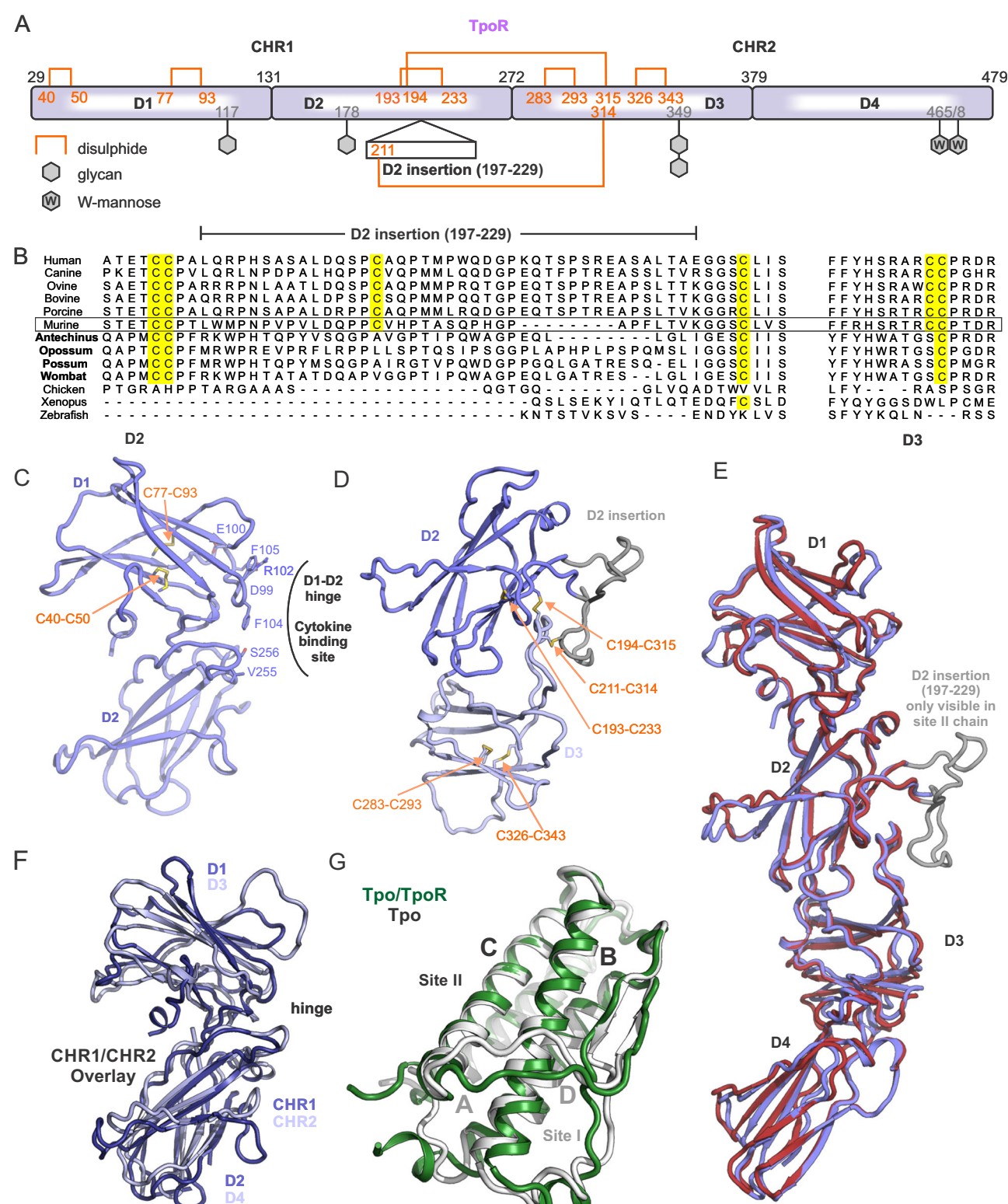

**Fig. 3 | Structural details of the Thrombopoietin receptor. A** Schematic of the domain architecture of murine TpoR. Disulphide bonds and glycosylation sites are shown. Significantly more density for glycans is observed at site 349 as compared to all other sites. TpoR contains an insertion in domain two (D2) as indicated. **B** Sequence alignment of the D2 insertion from different species. Marsupial TpoR (bold) homologues lack a single pair of cysteines. The alignment shown is flat-anchored against the human sequence and many species contain a much longer D2 insertion which has been omitted for clarity. **C** Cytokine binding homology region 1 (CHR1) showing key residues at site I. **D** The CHR1/CHR2 interface. Only domains 2 and 3 are shown. **E** Overlay of the two TpoR chains shown in cartoon representation. **F** Overlay of CHR1 and CHR2. **G** Overlay of mouse TpoR-bound Tpo (green) and human antibody-bound Tpo (white, PDB ID: 1V7N).

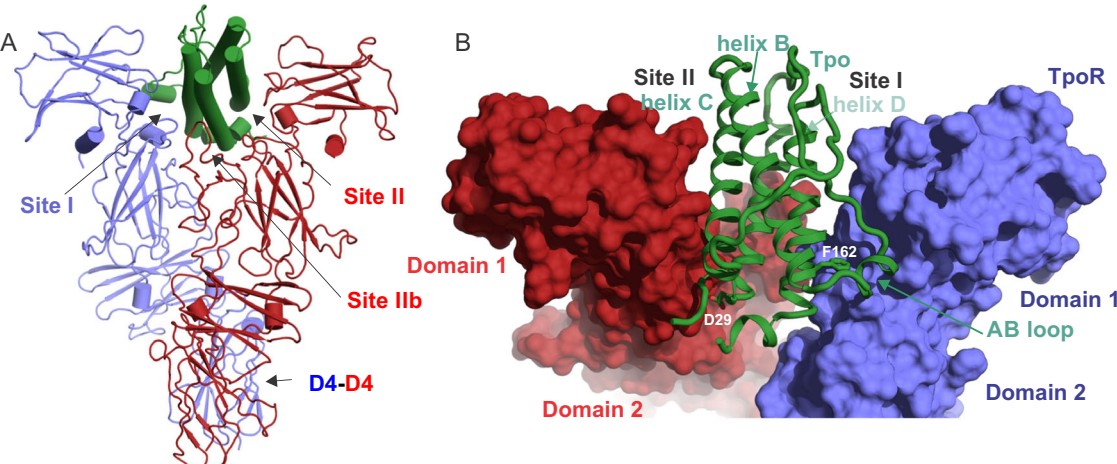

**Fig. 4 | The Tpo binding site. A** Ribbon diagram of the Tpo:TpoR structure with sites I, II, IIb and the Domain 4-Domain 4 (D4−D4) interactions indicated. **B** Closer view of the Tpo binding site with the receptor shown in surface representation.

Residues in Tpo that were subsequently mutated to disrupt the site I/II interactions are labelled in white.

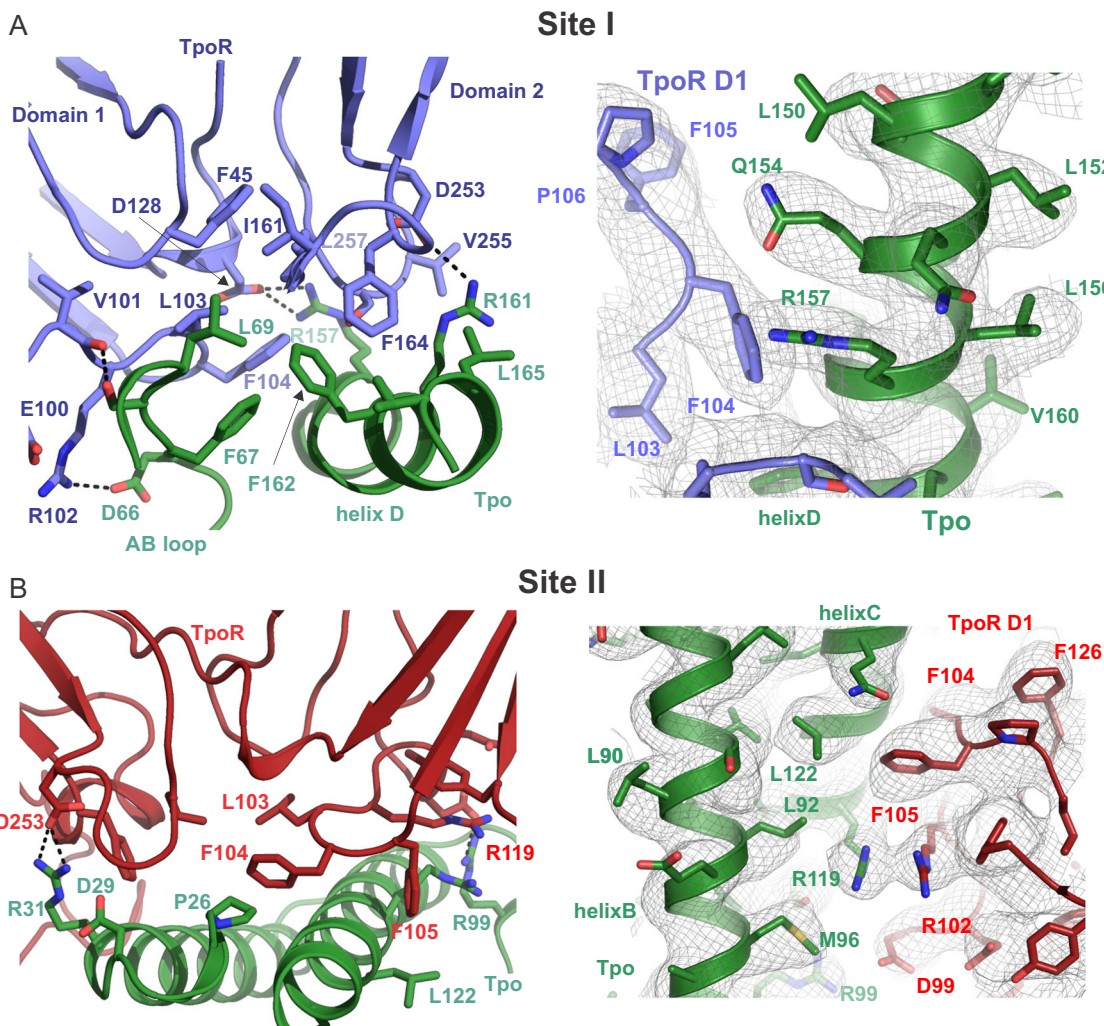

**Fig. 5 | Detailed view of Site I and Site II interaction interfaces.** The site I (**A**) and site II (**B**) interfaces are shown in cartoon and stick representation. Key polar contacts are illustrated with dotted lines. Representative density is shown on the right in both cases contoured at 6σ.

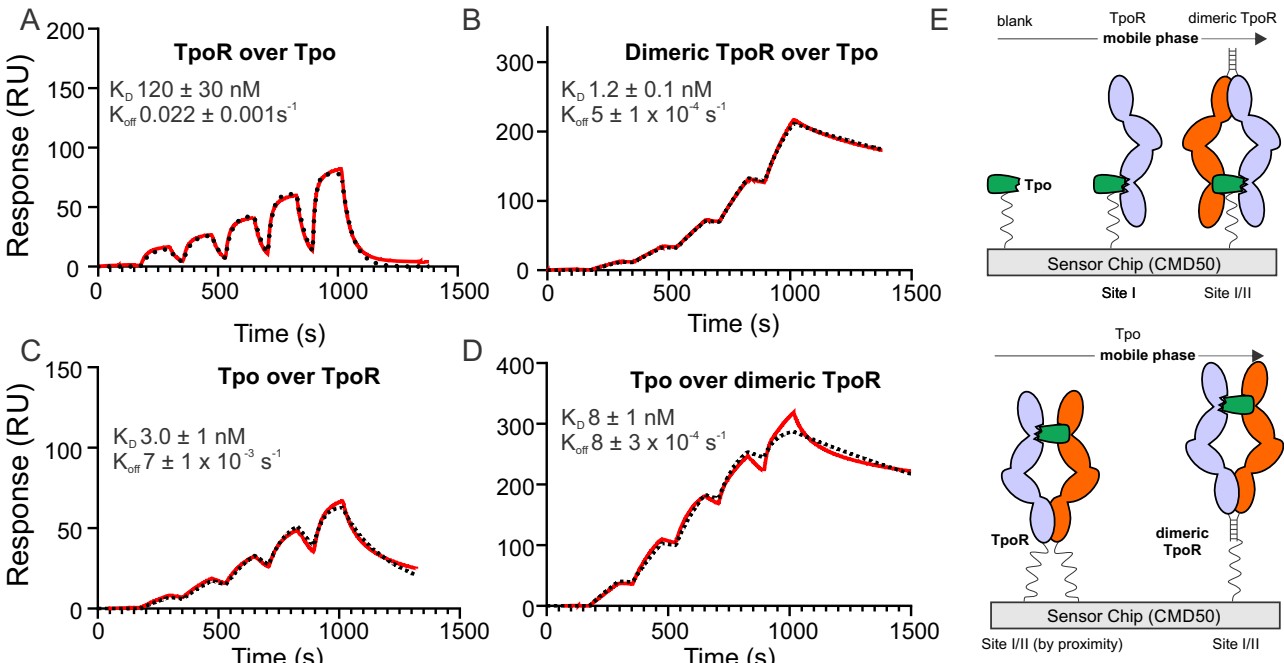

**Fig. 6 | Dimerisation of the Tpo receptor leads to high-affinity Tpo binding.** SPR analysis of monomeric and dimeric (leucine-zippered) TpoR. Upper panels show (**A**) monomeric and (**B**) dimeric receptors passed over immobilised Tpo. The affinity of the dimeric form is increased 100-fold. The lower panels show Tpo being passed over immobilised (**C**) monomeric and (**D**) dimeric receptor. In this case, the affinity is high regardless of the multimerisation state of the receptor. **E** Schematic of the interpretation of the data from (**A**–**D**). All data are from single cycle kinetics experiments fit to a single site binding model. Errors are standard deviations from triplicate experiments. A two-fold dilution series of analyte was injected over the chip surface, the highest concentration was **A** 250 nM, **B** 16 nM, **C** 15 nM, **D** 250 nM.

On the receptor, both site I and site II utilise TpoR F104. F104 is highly conserved across species (Supplementary Fig. 9) and was invariant in 450 closest homologues found by AlphaFold2[46]. Mutation of this residue (F104S) has also been observed clinically[47]. As predicted the F104S mutation completely abrogated binding to Tpo, whereas mutation of an adjacent residue (F105A), which is less conserved (Supplementary Fig. 9) and partially solvent exposed at both site I and II, had no effect (Supplementary Fig. 10).

### Identification of the romiplostim binding site

A potent peptide agonist of TpoR has been described[48], consisting of the sequence IEGPTLRQWLAARA, which binds with high affinity to a single copy of the receptor, and when dimerised, can bridge two copies of TpoR to initiate signalling. This was subsequently engineered as an Fc fusion (romiplostim), and is clinically used for the treatment of immune thrombocytopenia, but how it binds to TpoR is unknown. We generated a peptide version of romiplostim, using two copies of the above peptide sequence connected by a GG linker (IEGPTLRQW-LAARAGGIEGPTLRQWLAARA). This peptide bound both monomeric and dimeric TpoR in SPR experiments (Fig. 8A, B) and could induce differentiation of M1(TpoR) cells and growth of BaF3(TpoR) cells (Fig. 8C, D). As expected, the monomeric version of the peptide (IEGPTLRQWLAARA) had no biological effect.

Although SEC-MALS analysis clearly showed dimerisation of TpoR through the dimeric romiplostim peptide (Supplementary Fig. 13), as of yet we have been unable to obtain a high resolution cryo-EM structure of this complex. We therefore turned to AlphaFold2 as executed through ColabFold[46], to generate a model of human TpoR in complex with the peptide motif IEGPTLRQWLAARA. One of the resultant models indicated a binding site that overlaps with Tpo, as previously implied from cellular activity assays[49] and showed similarities with Tpo binding observed in our cryo-EM structure (Fig. 8E). The AlphaFold2 model indicated that P4, L6, W9 and L10 would form hydrophobic interactions with the receptor and is consistent with

limited mutagenesis of a cyclic version of the romiplostim sequence performed previously[50]. As shown in Fig. 8D, F mutating these residues led to a three-order-of-magnitude reduction in biological activity and no observable binding by SPR. We also tested the impact of mutating A12, predicted to be pointing away from the binding site, and this did not affect activity.

As the binding site of romiplostim on the receptor is predicted to overlap with that of Tpo, we used the TpoR mutants (F104S and F105A) we generated previously to investigate this. As shown in Fig. 8G, SPR assays indicated that, like Tpo, romiplostim cannot bind TpoR (F104S) whilst F105 is not required. Finally, competition SPR assays confirmed that Tpo competes for the romiplostim peptide binding site (Fig. 8H).

## Discussion

Thrombopoietin signalling is important in healthy blood cell development due to its roles in platelet production and maintenance of haematopoietic stem cells, and mutations in the receptor or associated regulators of the pathway manifest clinically as thrombocytopenia or myeloproliferative disease. The unique features of TpoR, namely the duplication of the CHR domain, and insertion of a large unstructured region within the second fibronectin(III) domain, mean it cannot necessarily be modelled from structures of the other family members. However, there has been a profusion of mutagenesis studies on both the cytokine and receptor and mapping these data onto our cryo-EM structure validates the observed interactions remarkably well.

As an example, Varghese et al. performed an analysis of TpoR extracellular domain mutants found in patients with congenital amegakaryocytic thrombocytopenia, as well as some residues predicted to be at the cytokine:receptor interface by homology modelling[51]. Their results show that a F105A mutant can still bind Tpo, but F45A, L103A, F104S, L257A and D128Y mutants cannot. This is entirely consistent with our structure and binding experiments, with all these residues clearly being important for site I high affinity interactions.

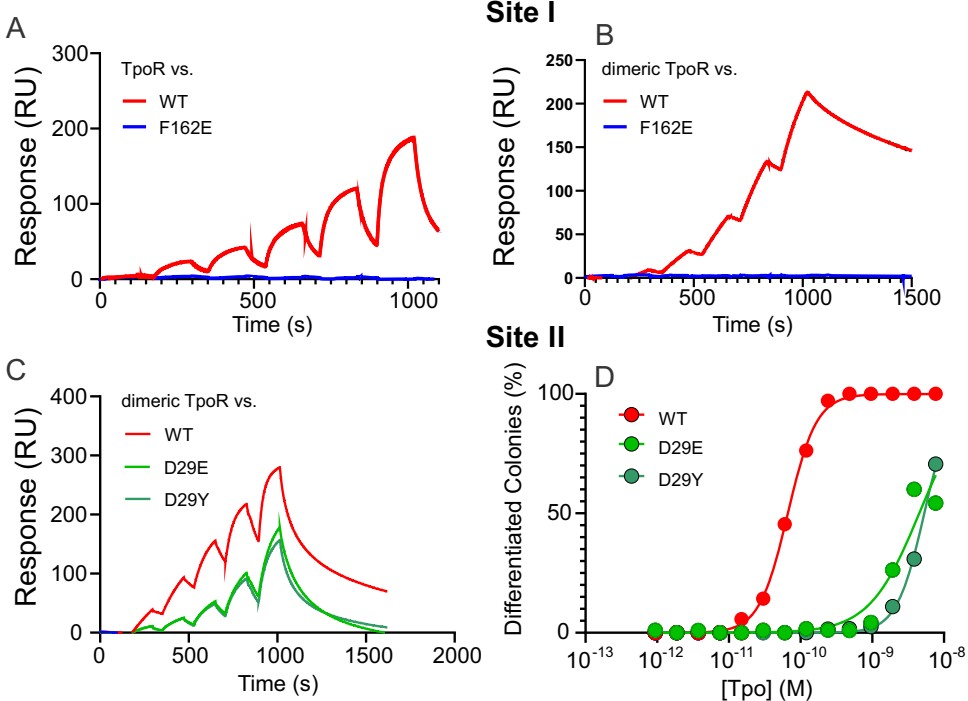

**Fig. 7 | Surface plasmon resonance and differentiation assays of Site I and Site II murine Tpo mutants.** Sensorgrams showing binding of TpoR (**A**) and dimeric TpoR (**B**) to immobilised Tpo (wild-type and F162E). **C** Sensorgrams showing binding of Tpo (wild-type and site II mutants) to immobilised dimeric TpoR (ectodomain) using SPR. D29E and D29Y both showed decreased affinity for receptor. The sensorgrams shown are representative of three independent experiments. A two-fold dilution series of cytokine was injected over the chip surface, the highest concentration was 50 nM. The affinity was calculated by fitting to a single-site binding model and was: WT 4.4 ± 0.1 nM; D29E 50 ± 10 nM; D29Y 23 ± 5 nM. **D** Site II mutants display an impaired ability to induce differentiation of M1(TpoR) cells.

Hao and Zhan performed an analysis of the residues of Tpo responsible for receptor binding, focussing on hydrophilic residues[35]. The results are in broad agreement with our structure. They highlight K159/R161 and D29/R31 as key residues in site I and site II, respectively. The loss of activity they observe for K159A is presumably a result of disrupting the CD-loop and, therefore loss of site I interactions, rather than direct interaction with the receptor. G158F, also located in site I, resulted in decreased potency. In general, a more subtle loss of activity was observed with site II mutants, except for the R31E charge swap mutation which led to a ~300-fold loss of potency.

## Structural comparison with other homodimeric cytokine receptors

TpoR has a duplicated ectodomain structure, consisting of two CHRs compared to shorter homodimeric receptors such as EpoR, PRLR and GHR, which contain only one. The structure indicates that the membrane-distal CHR binds cytokine using the hinge region between its two FnIII domains as is common amongst cytokine receptors. It is notable that both GHR and PRLR make extensive receptor:receptor contacts via their D2 domains (469 and 667 Å² buried surface area, respectively). In contrast, EpoR is positioned by its cytokine such that there is almost no D2-D2 contact. TpoR binds cytokine with a geometry that forces the two D2 domains apart (Fig. 9), ensuring that there is no receptor-receptor contact at this position. Instead, an interface is formed through D4-D4 interactions in the membrane proximal CHR. This D4-D4 interaction in the Tpo:TpoR complex (~110 Å² buried surface) is not as extensive as seen for PRL and GHR but may help position the transmembrane domains, and associated JAK-bound intracellular domains, in close proximity.

The unusual 33-residue insertion within D2 (Fig. 3) had much weaker associated density and only the main chain could be modelled. Its role remains unclear. Although it contacted cytokine at one position, it is unlikely to add much affinity to that interaction. This insertion is very variable in length in mammalian and avian species and is not present in zebrafish or xenopus homologues. TpoR exists largely as a monomer on the cell surface, with some transient homodimerization[16]. It is tempting to speculate that the steric bulk of this large flexible loop acts to limit unwanted homodimerization. It is known that deletion of CHR1 (including the D2 insertion) leads to cytokine-independent signalling[14]. The absence of the D2 loop may perturb the dynamic monomer-homodimer equilibrium and sustain the interaction long enough to enable JAK autophosphorylation. This is seen with TpoR signalling through JAK2-V617F, which appears to prolong the length of the cytokine-independent homodimerization events and allow JAK activation to occur[16].

Another question that remains is why TpoR has evolved a duplication of the CHR. The hinge region in these modules is commonly the site of cytokine engagement. In TpoR, this site in the membrane proximal CHR is glycosylated (N349) and here we observe the most density for any glycosylation site in the receptor. This argues against it having a role in recruiting a cytokine and glycosylation may in fact be necessary to prevent inadvertent binding by other cytokines or cytokine-like molecules.

## Site I and site II

Structural details of the site I and II interfaces indicate a lower buried surface area compared to other homodimeric receptors (Tpo: 870 Å², 870 Å²; GHR: 1254 Å², 816 Å²; PRLR: 1273 Å², 973 Å²; EPO: 975 Å², 696 Å²). In all receptors there is a significant difference between the site I and site II affinities. However, the affinity of site I of TpoR that we infer from SPR analyses ($K_D$ 100 nM) is notably weaker than other homodimeric receptors such as EPOR, GHR and PRLR (Site I $K_D$:1–30 nM)[52,53]. Likewise, the site II interaction could not be quantified or even observed (by gel filtration, SEC-MALS, SPR or mass-photometry) unless

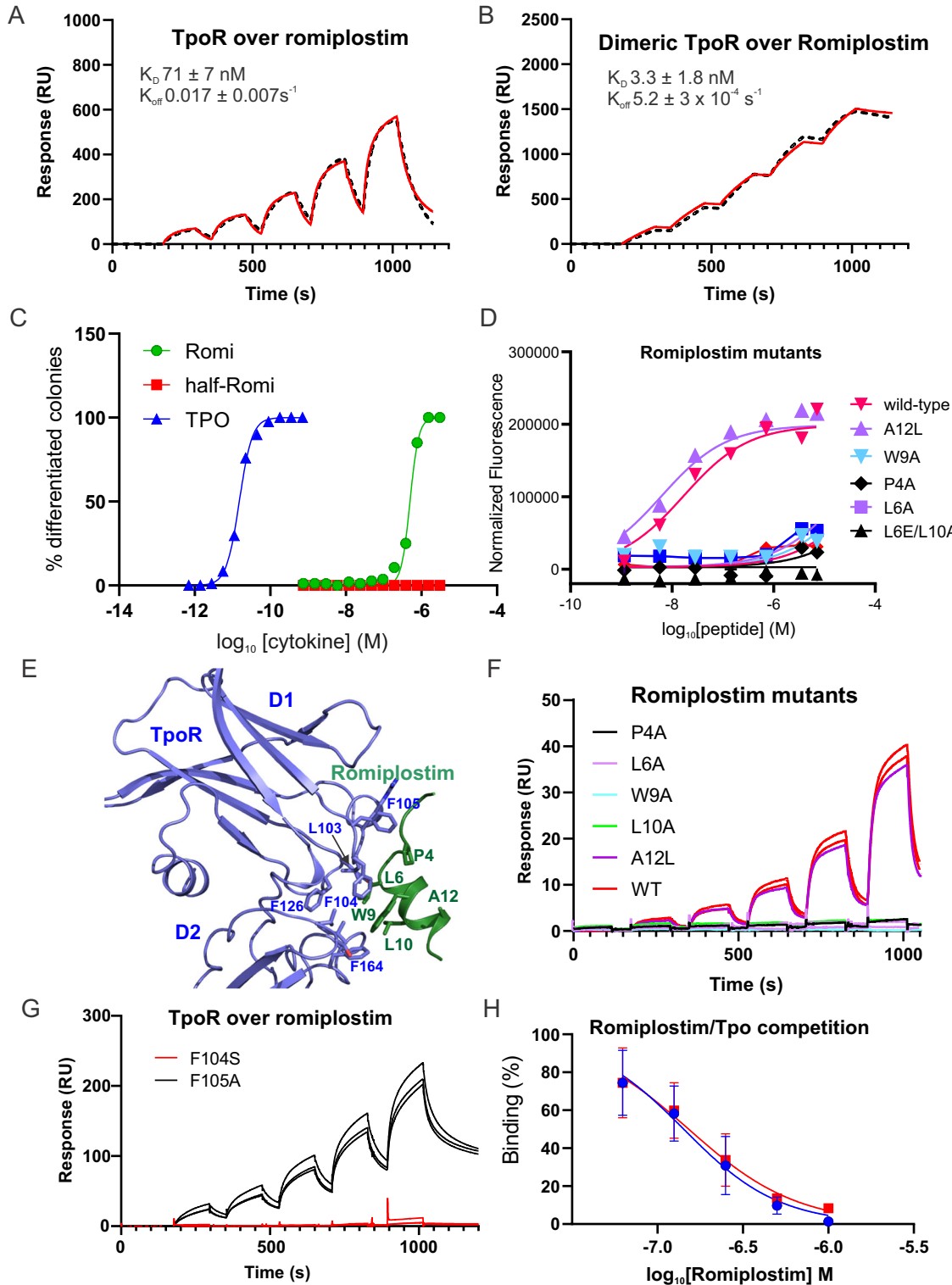

**Fig. 8 | Interaction of romiplostim with TpoR.** SPR analysis of (**A**) monomeric and (**B**) dimeric (leucine-zippered) TpoR passed over immobilised romiplostim peptide. Errors are standard deviation from triplicate experiments. A two-fold dilution series of receptor was injected over the chip surface, the highest concentration was 100 nM. **C** Romiplostim induces differentiation of M1(TpoR) cells. M1(TpoR) cells were plated in semi-solid agar and exposed to Tpo. The results shown are representative of two independent experiments. **D** Growth of Ba/F3(TpoR) cells in response to dimeric romiplostim peptide and variants. Results shown are the average and standard deviation of duplicate experiments. **E** AlphaFold2 prediction of the romiplostim peptide (green) in complex with TpoR (blue). **F** SPR analysis of

TpoR passed over immobilised monomeric-romiplostim and mutants thereof. Two independent experiments are shown overlaid. A two-fold dilution series of TpoR was injected over the chip surface. 100 nM was the top concentration. **G** SPR analysis of TpoR mutants passed over immobilised romiplostim peptide. F104S but not F105A abolishes binding to romiplostim. Data shown are technical triplicates and are representative of three independent experiments. **H** Romiplostim blocks TpoR binding to immobilised Tpo: 125 nM TpoR was passed over immobilised Tpo in the presence of increasing concentrations of romiplostim. The response was normalised to the no romiplostim control. Data shown are two independent experiments (red, blue) with error bars representing standard deviation of technical triplicates.

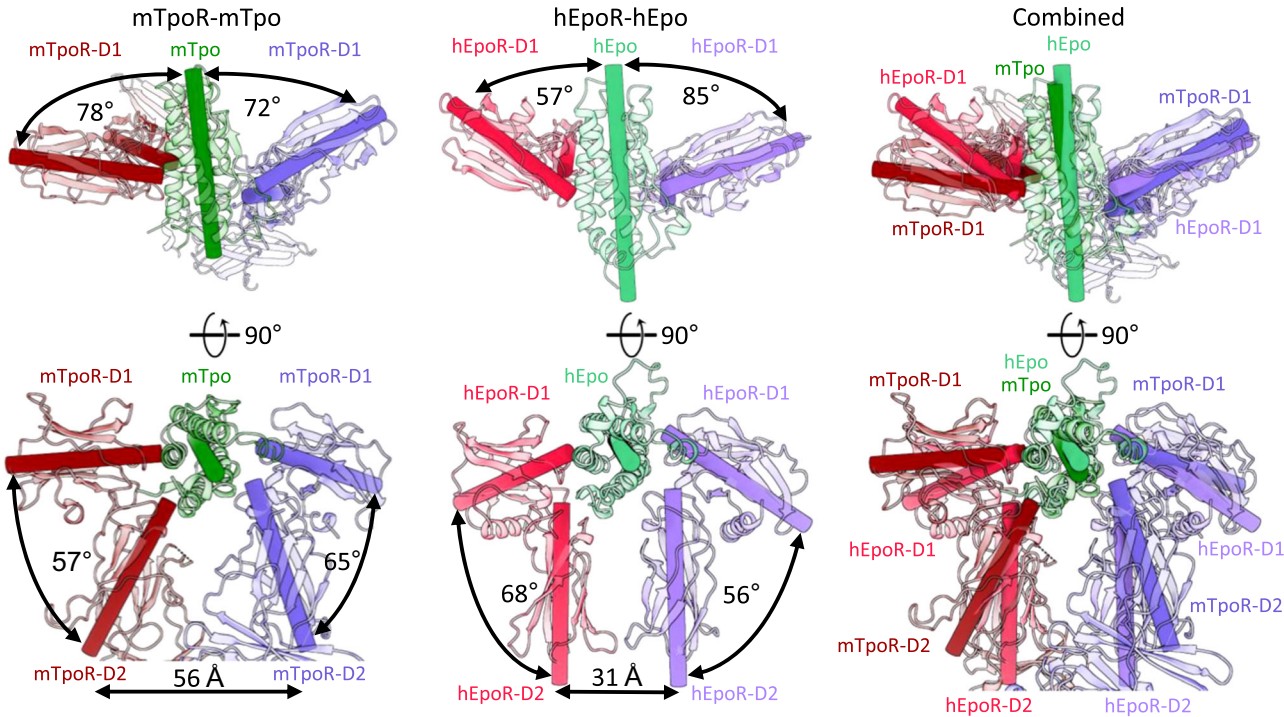

**Fig. 9 | Comparison of receptor D1-D2 positions relative to cytokine between murine Tpo:TpoR and human Epo:EpoR.** Angles and distances were measured in ChimeraX[85] using axes placed along the centre of mass of each domain. Epo:EpoR PDB ID: 1EER[43].

the receptor was first dimerised. Our attempts to observe a 1:2 Tpo:TpoR complex indicate a $K_D \gg 10\,\mu M$ for site II, similar to that observed for PRLR ($30\,\mu M$[54]) but weaker than that of the EPO:EPOR and GH:GHR interactions ($1\,\mu M$, 3 nM respectively[52,53]). It is interesting that such a discrepancy between sites I and II exists as the buried surface and shape complementarity scores were similar for both sites. The major difference was in the number of predicted hydrogen bonds which was significantly higher in site I. There have been very few clinical mutations identified in Tpo that give rise to thrombocytopenia. However, those that have (R119C[55], R38C[56], R99W[57]) are all located in site II, highlighting the importance of this interaction.

Although site I was of modest affinity and site II/III was unmeasurable, the affinity of Tpo for the dimerised receptor was much higher ($K_D$ ~1 nM), similar to that reported previously[58]. The off-rate was particularly slow and this led to a very stable complex (half-life >20 min). Despite the higher affinity we were intrigued by the fact that it was still an order of magnitude weaker than its biological activity as measured here in both differentiation and growth assays ($EC_{50}$ ~40 pM) and previously[51]. Previous studies have shown that certain conformations of the TpoR do not support full signalling throughput[59] and that human TpoR is more restricted than murine TpoR in this regard[60] due largely to the presence of His499 in the human protein. We constructed a limited series of insertion mutants in the leucine zipper to determine whether changes in the helical register of the junction between the ectodomain and the leucine-zipper would affect binding of the cytokine. Although we observed impaired binding by one such mutant, it is unclear whether that is due to a specific conformation or is merely due to that particular construct placing the ectodomains furthest apart (Supplementary Fig. 10). There is enough flexibility, in vitro, between the ectodomain and the leucine zipper (or the transmembrane domain[33]) such that no density for these regions, or several residues that precede it, were visible in either TpoR structure[33]. Therefore, any conformational change transmitted from the ectodomain to the transmembrane domain upon cytokine binding seems unlikely in solution. In vivo, it

is possible that there is a subtle re-orientation of the receptor on the cell surface or the membrane itself that could play a role in signal transduction, for example by driving the D4:D4 interaction. Indeed, there is a concentration of arginines on the membrane proximal end of D4 and a tryptophan at the membrane junction, which could aid this. Such an interaction occurs immediately downstream of the transmembrane domain, in the juxtamembrane sequence and is important for preventing aberrant activation in the absence of cytokine[28,61,62]. Another possible explanation of the low $EC_{50}$ in vivo is that that the maximum biological response by Tpo is generated by only fractional receptor occupancy. Indeed, a recent study observed a 1 nM $EC_{50}$, similar to the in vitro $K_D$, when STAT3 phosphorylation (rather than biological outcome) was used as a read-out[16].

Comparison of our murine Tpo:TpoR structure with that of the human Tpo:TpoR complex[33], both of which were obtained at comparable resolution, reveals that they are remarkably similar (RMDS 2.0Å for C-alpha over the entire complex, Supplementary Fig. 14), despite quite different approaches to obtaining the structure. In this study we used Expi293 cells to make TpoR and showed that the glycans were mature glycoforms. Tsutsumi et al.[33] used 293GnTI- cells, expected to produce immature, high-mannose glycans[63]. While our approach used a Leucine-zippered construct to generate dimeric recombinant ectodomain, Tsutsumi et al.[33] used the full-length receptor extracted into detergent micelles. Neither the leucine zipper nor the transmembrane domain were visible in either structure. Both studies used only the cytokine bundle of Tpo, with the majority of the C-terminal unstructured region removed; however, Tsutsumi et al. used a cross-linking agent to stabilise the cytokine:receptor complex, whereas in our study we were able to isolate the complex without the need for cross-linking.

The mouse and human receptors share 78% sequence identity (83% similarity) within their ectodomain. These sequence differences are located uniformly throughout the receptor and four of them are within 4.5 Å of site I and/or site II. Of these, mouse TpoR R70 (Proline in human TpoR) forms a hydrogen bond with Tpo Q126, and D99

(Glutamate in human TpoR) hydrogen bonds with Tpo R99. Equivalent interactions are not present in the human Tpo:TpoR structure. The biggest difference between the two species is the D2 insertion. The human insertion is eight residues longer and shares only 42% homology with the mouse sequence. However, this insertion was not resolved in the human structure and was only weakly resolved in one chain of the structure presented here, where it was seen to contact cytokine, forming site IIb. We observe that the insertion contacts cytokine at one end and D3 (to which it is disulphide bonded) at the other. The latter disulphide was not resolved in the human Tpo:TpoR structure. Interestingly, there is only a single cysteine not engaged in a disulphide bond, C413, in D4. This cysteine, which is conserved in placental mammals but not in other species, is located 13 Å from its counterpart in the other receptor chain. Even though flexibility is seen in the legs of the receptor that can bring the two D4 domains slightly closer together (Supplementary Movie 1), it seems unlikely that they would disulphide bond in vivo.

Biochemically, there are differences in the SPR analyses of site I and site II in the current manuscript and that describing the human complex[33]. We measure a $K_D$ of 100 nM between Tpo and a single chain of TpoR when the cytokine is immobilised on the chip surface and have interpreted this as the site I interaction. When the reciprocal experiment is performed (receptor is immobilised) the resultant affinity is 100× higher ($K_D$ 1 nM) which we have interpreted as proximity-induced binding of the cytokine to *both site I and site II from nearby chains on the chip surface*. Tsutsumi et al.[33] measured the same affinity under these conditions but infer that it is due to site I only. In both studies, the site II affinity was too low to be quantified.

Formation of the active dimeric TpoR receptor appears to depend on multiple contact points: receptors are bridged by cytokine, and contact each other at D4, the transmembrane helix and JAK pseudo-kinase domain[16,64]. The propensity of the receptor to dimerise in the absence of cytokine is higher for mouse than for human, and it has been shown that removal of the ectodomain of mouse TpoR (but not human) results in a further increase in receptor dimerisation[65]. Given the similarity in the binding sites and overall conformation of the mouse and human Tpo:TpoR ectodomain complexes (Supplementary Fig. 14), it seems likely that the propensity to dimerise or not is a result of the properties of the transmembrane region itself, rather than differences in the ectodomain. In particular His 499, which is Leu in mouse TpoR, has been shown to limit the formation of pre-formed dimers in the human protein by altering the structure of the transmembrane domain and breaking the helical nature[65], which is presumably restored in the active conformation when cytokine induces dimerisation. Obtaining a structure of the full-length receptor in the presence of cytokine and JAK presents the next structural biology challenge in understanding the details of this system.

### Interaction with romiplostim

Romiplostim is an Fc-dimerised peptide of sequence IEGPTLRQWLAARA. AlphaFold2 predicts that this peptide forms an amphipathic helix whose hydrophobic side binds to the Tpo-binding site on the receptor. Our data are consistent with this prediction. We show that this peptide competes for Tpo binding to the receptor, requires F104 (but not F105) for binding and that the dimeric version can recruit two TpoR chains and induce both cell growth and differentiation. Romiplostim has shown benefit to patients with hereditary and drug-induced thrombocytopenias[66–68] as well as bone marrow disorders such as aplastic anaemia[69], however its efficacy is currently not sufficient for FDA approval for those indications and it is only approved for use in immune thrombocytopenia. The structure of the Tpo:TpoR complex, along with modelling of the romiplostim:TpoR interaction, opens the way for rational design of improved agonists with important therapeutic applications.

## Methods

### Expression and purification of Murine TpoR$_{1-479}$ and TpoR$_{1-482}$-LeuZ

A construct expressing a fusion protein consisting of residues 1-479 of murine TpoR, followed by a TEV cleavage site and the Fc domain of hIgG1 under a CMV promoter (Supplementary Table 1), was used to transiently transfect Expi293F™ cells (Gibco). Cells were cultured to a density of $2.5 \times 10^6$ cells per mL in polycarbonate Erlenmeyer flasks with shaking at 130 rpm, 37 °C and 8% CO$_2$. DNA was diluted in DPBS (Gibco) and mixed with PEI 25k (Polysciences) at a ratio of 3:1 PEI:DNA and incubated for 20 minutes at room temperature before being added to cells at a final concentration of 1 mg DNA per 1 L of culture. Cells were harvested by centrifugation 6–7 days post-transfection and the supernatant collected and stored at 4 °C. TpoR$_{1-482}$-LeuZ and the four leucine zipper length extensions were generated in the same way (Supplementary Table 1). The leucine zipper sequence was that of GCN4, *Saccharomyces cerevisiae*: RMKQLEDKVEELLSKNYHLENEV ARLKKLVGERTG.

Fc-fusion proteins were captured from the supernatant using Protein G Sepharose 4 Fast Flow resin (Cytiva). Supernatant was incubated with resin overnight at 4 °C with rotation. Beads were recovered by pouring into an Econopac chromatography column (BioRad) and washed twice with 20 column volumes of Tris-buffered saline (TBS), pH 7.5. Protein was eluted from the resin using a solution of 100 μg/mL TEV in TBS (pH 7.5) with 2 mM βME for 2 h at room temperature on a rotating wheel. Eluted protein was concentrated before being applied to a Superdex 200 size-exclusion column (Cytiva) equilibrated in TBS, pH 7.5. Purest fractions as analysed by both reducing and non-reducing SDS-PAGE were pooled, snap-frozen and stored at −80 °C.

### Expression and purification of Tpo

Tpo was cloned into pcDNA3.0, as a fusion of the mIL3 secretion signal(sequence), N-terminal FLAG-tag followed by residues 22 to 356 of mouse Tpo (Supplementary Table 1). The construct was used to transfect FreeStyle™ 293-F Cells (Gibco) at a density of $1 \times 10^6$ cells per mL. Transfection was performed as for TpoR (above), and cells were grown for 5-6 days post-transfection. 293 F media was supplemented with GlutaMAX™ (Gibco), 0.2 mM butyric acid (Sigma−Aldrich) and 5 g/L lupin (Solabia), 48 h after transfection.

Secreted recombinant Tpo was purified using an ANTI-FLAG® M2 Affinity Gel (Sigma-Aldrich). The protein was eluted with 150 μg/mL FLAG peptide in TBS (pH 7.5) and subsequently concentrated and applied to a Superdex 200 size exclusion column (Cytiva) equilibrated in TBS (pH 7.5). Non-reducing SDS-PAGE analysis indicated that the major peak was misfolded aggregate, but a small shoulder adjacent to the major peak contained pure, monomeric, full-length Tpo. The N-terminal domain (Tpo$_N$, 22–184) was generated by mutagenesis, inserting a stop codon in place of residue 185 (Supplementary Table 1). It was expressed and purified using the same protocol as for full length Tpo.

### Surface plasmon resonance

All SPR experiments were run on a Biacore 8k (Cytiva) instrument using the Biacore 8k control software (v. 3.0.12.15655, Cytiva). Proteins at a concentration of 1 μg/ml (Tpo, TpoR) in 10 mM sodium acetate (pH 4.6) were directly coupled to a CM5 chip using EDC/NHS (N-ethyl-N′-(3-(dimethylamino)propyl)carbodiimide/N-hydroxysuccinimide) according to a standard protocol (Cytiva). Romiplostim peptides were synthesised with an N-terminal biotin moiety and coupled to streptavidin chips (SA, Cytiva or SAD 200 M Xantec). In every case, flow cell one was unmodified and used as a negative control. For binding assays, analyte proteins were dissolved in HBS-EP and titrated over the chip surface in concentrations varying from 0 to 1 μM. In general, single-cycle kinetics

experiments were performed due to the slow off-rate of the majority of complexes studied. These experiments consisted of 6 successive injections of analyte of increasing concentration (e.g. 0, 31.25, 62.5, 125, 250, 500 nM) after a negative control cycle using running buffer only. For analysis, the sensorgram from the negative control cycle was subtracted from the experimental cycle to yield the final result which was analysed using Biacore Insight evaluation software v. 3.0.12.1565 (Cytiva) using the 1:1 binding kinetics module.

## M1(TpoR) differentiation assay

M1(TpoR) cells, generated by retroviral incorporation of murine TpoR into M1 cells (WEHI), were grown in DME with 10% (v/v) FBS, counted using a hemocytometer, centrifuged at $425\,g$ for 5 min at 4 °C and then resuspended at the desired cell titre. A cocktail containing 30% (v/v) modified double strength DME, 20% (v/v) FBS (Sigma-Aldrich) and 50% of 0.6% (w/v) Bacto Agar (pre-warmed to 42 °C) were mixed with M1 cells (final concentration 300–400 cells/mL) and 1 mL aliquots added to Petri dishes with 0.1 mL of Tpo or a saline control. After settling at room temperature for 15 min, cultures were incubated at 37 °C and 10% $CO_2$ in a humidified incubator for 7 days and then differentiated and undifferentiated colonies were counted using an Olympus dissection microscope. Differentiated colonies are not tightly formed and have a halo of migrating cells around their periphery[36]. At high Tpo concentrations colony formation is completely suppressed as the cells undergo terminal differentiation and apoptosis.

## Ba/F3(TpoR) growth assay

Ba/F3 (TpoR) cells, generated by retroviral incorporation of murine TpoR into Ba/F3 cells (WEHI), were grown in RPMI 1640 supplemented with 10% (v/v) FBS and WEHI 3B media (Sigma-Aldrich) at 37 °C with 10% $CO_2$. The cells were washed three times with RPMI/FBS to remove traces of WEHI 3B and then 200 µl added to wells of a 96-well plate in the presence and absence of romiplostim. The cells were incubated for 4 days at 37 °C, 10% $CO_2$ and then cell growth was measured using a CellTitre-Glo assay (Promega).

## Mass photometry

MP experiments were performed on a Refeyn TwoMP (Refeyn Ltd.). Cytokine and receptor were diluted into MES-buffered saline at pH 6 and mixed for 10 min prior to recording. A 3 mm six-well sample cassette was placed onto a clean glass slide and 10 µL of buffer was placed into the well for focusing. Following this, 10 µL of sample was added and mixed by pipetting. Movies were recorded for 60 s at a frame rate of 475 Hz using AcquireMP (v.2023 1.1.0, Refeyn Ltd). Frames were binned into groups of ten. Size estimates were determined by calibration using a mixture of apoferritin, thyroglobulin, catalase and BSA and data processed using DiscoverMP (v. 2023 R.1.2, Refeyn Ltd).

## Sample preparation for LC-MS analysis

50 µg of purified TpoR was solubilized in 4% SDS, 100 mM Tris pH 8.5 by boiling for 10 min at 95 °C then prepared for digestion using Micro S-traps (Protifi, USA) according to the manufacturer's instructions. Briefly, samples were reduced with 10 mM DTT for 10 min at 95 °C and then alkylated with 40 mM IAA in the dark for 1 h. Samples were then split into two aliquots, acidified to 1.2% phosphoric acid and diluted with seven volumes of S-trap wash buffer (90% methanol, 100 mM Tetraethylammonium bromide pH 7.1) before being loaded onto S-traps and washed three times with S-trap wash buffer. Each aliquot was digested with either 5 µg of Solu-Trypsin (Sigma) or 5 µg of GluC (Promega) overnight at 37 °C before being collected by centrifugation with washes of 100 mM Tetraethylammonium bromide, followed by 0.2% formic acid, followed by 0.2% formic acid/50% acetonitrile. Samples were dried down and further cleaned up using C18 Stage[70,71] tips to ensure the removal of any particulate matter.

## Reverse phase liquid chromatography–mass spectrometry

C18 enriched digests were re-suspended in Buffer A* (2% acetonitrile, 0.01% trifluoroacetic acid) and separated using a two-column chromatography setup composed of a PepMap100 $C_{18}$ 20-mm by 75-µm trap (Thermo Fisher Scientific) and a PepMap $C_{18}$ 500-mm by 75-µm analytical column (Thermo Fisher Scientific) using a Dionex Ultimate 3000 UPLC (Thermo Fisher Scientific). Samples were concentrated onto the trap column at 5 µl/min for 6 min with Buffer A (0.1% formic acid, 2% DMSO) and then infused into an Orbitrap Lumos™ Mass Spectrometer (Thermo Fisher Scientific) at 300 nl/min via the analytical columns. Peptides were separated by altering the buffer composition from 3% Buffer B (0.1% formic acid, 77.9% acetonitrile, 2% DMSO) to 28% B over 120 min, then from 23% B to 40% B over 9 min and then from 40% B to 80% B over 2 min. The composition was held at 80% B for 2 min before being returned to 3% B for 2 min. The Orbitrap Lumos™ was operated in a data-dependent mode, automatically switching between the acquisition of a single Orbitrap MS scan and 3 s of scouting Orbitrap MS/MS HCD scans to identify potential glycopeptides. For each digest samples were analysed using both a standard MS1 ranges (350–2000 m/z, maximal injection time of 118 ms, an Automated Gain Control (AGC) set to a maximum of 100% and a resolution of 60k) or high m/z focused MS1 range (600–2000 m/z, maximal injection time of 118 ms, an AGC set to a maximum of 100% and a resolution of 60k) to aid in the identification of low abundance glycoforms. Scouting HCD scans (NCE of 30%, a maximal injection time of 60 ms, a AGC of 250% and a resolution of 30k) containing HexNAc associated oxonium ions (204.0867; 138.0545 and 366.1396 m/z) triggered two additional product-dependent MS/MS scans[72] of potential glycopeptides; a Orbitrap EThcD scan (NCE 15%, maximal injection time of 250 ms, a AGC of 300% and a resolution of 30k with the extended mass range setting used to improve the detection of high mass glycopeptide fragment ions[73]); and a stepped collision energy HCD scan (using NCE 28;35 and 40%, maximal injection time of 250 ms, a AGC of 300% and a resolution of 30k).

## Glycopeptide analysis

TpoR digests were searched with FragPipe version 20[74–78] using the "glyco-N-HCD" workflow with either a Trypsin or GluC enzyme specificity allowing carbamidomethyl as a fixed modification of cysteine in addition to oxidation of methionine and N-terminal acetylation. To identify additional modifications an "Open" search was undertaken on the trypsin digested LC-MS data. Searches were performed against a database containing the predicted TpoR sequence as well as potential contaminate sequences allowing a 1% FDR. The resulting "psm.tsv" files for each digest were combined using R retaining only glycopeptides with a MSfragger Hyperscore >20. Visualisation of proteomic data was undertaken using R (version 4.2.1) and the tidyverse[79] collection of packages. Glycoforms were assigned according to the guidelines of ref. 80 with HexNAc(2)Hex(1–10) classified as M1 to M10; HexNAc(3)Hex(5–6)X or HexNAc(3)Fuc(1)X assigned as Hybrid-type glycans while Complex-type glycans were defined according to the level of fucosylation and processed antenna with HexNAc(3)Hex(3-4)X assigned as A1; HexNAc(4)X as A2/A1B; HexNAc(5)X as A3/A2B; and HexNAc(6)X as A4/A3B. C-glycosylation events identified within the "Open" search were annotated with the Interactive Peptide Spectral Annotator[81] (http://www.interactivepeptidespectralannotator.com/PeptideAnnotator.html). The mass spectrometry data associated with this analysis has been deposited to the ProteomeXchange Consortium via the PRIDE partner repository with the dataset identifier PXD046926. Data are also provided in Supplementary Data 1.

## Cryo-EM specimen preparation and data collection

Purified $TpoR_{1-479}$-LeuZ and $Tpo_N$ were combined in a molar ratio of 1:1.5, incubated for 5 min at room temperature then purified by size-exclusion chromatography using a Superdex 200 10/300 Increase

column (Cytiva) equilibrated in TBS (pH 7.5). The resultant peaks were analysed by SDS-PAGE and SEC-MALS, and those corresponding to the complex were pooled and concentrated to 0.55 mg/mL before snap freezing in liquid nitrogen and storage at −80 °C.

Initial attempts at cryo-EM of Tpo:TpoR indicated that preferential orientation limited the views of the complex and high-resolution reconstruction. Addition of 0.005% (w/v) cetrimonium bromide (VitroEase Buffer Screening Kit, ThermoFisher Scientific) shortly before plunge freezing resulted in sufficient orientations for a high-resolution reconstruction. Briefly, a 3 μL aliquot of TpoR$_{1-479}$-LeuZ:Tpo$_N$ complex (0.55 mg/mL) with added CTAB was applied to a glow-discharged R1.2/1.3 holey carbon grid (Quantifoil Micro Tools GmbH, Germany) and plunge-frozen in liquid ethane using a Vitrobot Mark IV (ThermoFisher Scientific) with a blot time of 4 s, blot force of 6 and 0 s drain time. Grids were transferred under liquid nitrogen to a Titan Krios G4 transmission EM (Thermo Fisher Scientific) operated at 300 keV and set for parallel illumination. A dataset of 6,130 movies was recorded using EPU 2 (FEI) on a K3 Summit direct electron detector (Gatan Inc., USA) with zero-loss energy filtering at a calibrated magnification of 105,000 X (Table 1).

### Cryo-EM data processing and 3D reconstruction
Reconstruction of the Tpo:TpoR structure was performed using cryoSPARC (v.4.3.0) (summarised in Supplementary Fig. 2)[82]. Movies were aligned using patch motion correction and defocus values were estimated using patch contrast transfer function (CTF) estimation. Images with significant astigmatism, ice contamination, or drift were removed resulting in 5297 micrographs. Templates for picking were generated using a gaussian based picking approach, followed by 2D classification and selection of high-quality 2D classes for template picking. An initial 6,480,599 picks were reduced to 2,353,523 picks by adjusting the power and normalised cross correlation (NCC) thresholds. Following 2D classification which retained 1,924,121 particles (81.8%), a subset of the data (885,985 particles) was used to create an initial model via multiple rounds of heterogenous refinement and one round of non-uniform refinement[83]. This 3D reference was used to perform local motion correction and CTF estimation on all particles. The motion corrected particles were then subjected to three rounds of heterogenous refinement followed by a single round of non-uniform refinement, resulting in an intermediate reconstruction at 3.7 Å. Further 2D classification, another round of motion correction and local CTF estimation, as well as estimation of beam tilt, improved the resolution for the final reconstruction to 3.6 Å. To resolve the density of domains 3 and 4, we used 3DFlex[40] and performed local sharpening using DeepEMhancer (v. 0.15)[84].

### Model building and refinement
An initial model of Tpo:TpoR was produced using AlphaFold2 on Google CoLab[46] and fitted into the maps using rigid body fits of each domain in ChimeraX (v. 1.6.1)[85]. Modelling was performed in Coot (v. 0.9.8.3)[86] and ISOLDE (v. 1.6.0)[87] and the models were refined in real space with the phenix.real_space_refine programme (PHENIX v. 1.20.1_4487)[88] using secondary structure restraints. The geometry and quality of the models were evaluated using a combination of MolProbity[89] and PHENIX. Visualisation and analysis of the models and maps were performed using ChimeraX[85] and PyMOL (Schrödinger, LLC, v. 2.5.0)[90].

### Reporting summary
Further information on research design is available in the Nature Portfolio Reporting Summary linked to this article.

## Data availability
The data that support this study are available from the corresponding authors upon request. The structural data (atomic coordinates and cryo-EM density maps) generated in this study have been deposited in the Protein Data Bank (PDB) and Electron Microscopy Data Bank (EMDB) databases under accession code 8U18 (murine Tpo:TpoR complex) and EMD-41805 (murine Tpo:TpoR complex). The structures used for comparison in this study are available in the Protein Data Bank under the following accession codes: 1V7N (human Tpo); 1EER (human Epo:EpoR complex); 3HHR (human GH:GHR complex); 3NPZ (human PRL:rat PRLR); 8G04 (human Tpo:TpoR complex). All mass spectrometry data (RAW files, FragPipe outputs, Rmarkdown scripts, and input tables) have been deposited into the PRIDE ProteomeXchange repository[91] with the data set identifier: PXD046926. All other data generated in this study are provided in the Supplementary Information. The source data underlying Figs. 1, 6, 7, 8a–d, 8f–h and Supplementary Fig. 1f-g, 10a-b, 13 are provided as a Source Data file. Source data are provided with this paper.

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

## Acknowledgements

We thank the staff at the Ian Holmes Imaging Centre, Bio21 Institute, for access to and assistance with the Titan Krios microscope used for data collection. This research was undertaken with the assistance of Milton HPC/Virtual Machines, supported by WEHI. This work was supported by philanthropic donations through the WEHI accelerator fund and via a Medical Research Future Fund (MRFF) grant MRF2008972. K.S.L. was supported by a Walter and Eliza Hall Handman PhD scholarship, J.M.H. is supported by an NHMRC Investigator Grant (2008096) and is a member of the Australian Research Council Industrial Transformation Training Centre for Cryo-Electron Microscopy of Membrane Proteins for Drug Discovery (IC200100052). N.E.S. is supported by an Australian Research Council Future Fellowship (FT200100270), an Australian Research Council Discovery Project Grant (DP210100362) and a National Health and Medical Research Council Ideas Grant (2018980). We thank the Melbourne Mass Spectrometry and Proteomics Facility of the Bio21 Molecular Science and Biotechnology Institute for access to MS instrumentation. The contents of this published material are solely the responsibility of the individual authors and do not reflect the views of the NHMRC or funding partners.

## Author contributions

N.J.K. and J.J.B. designed the study. N.J.K., J.I., K.S.L. and T.S. performed protein purifications. D.S., C.T., N.A.N. and K.B. performed cell-based experiments. K.S.L. and J.M.H. performed cryo-EM experiments and reconstruction. A.P.L. assisted with cryo-EM data collection. K.S.L., J.M.H., J.J.B. and N.J.K., built and analysed the Tpo:TpoR model. J.J.B., T.A.D., T.S. and K.S.L. performed biochemical analyses. L.J. and N.E.S. performed mass spectrometry analyses. J.M.H., J.J.B. and N.J.K. wrote the manuscript.

## Competing interests

The authors declare no competing interests.
