## [Peer Review File · Nature Communications]

Cryo-EM structure of the extracellular domain of murine Thrombopoietin Receptor in complex with ThrombopoietinREVIEWER COMMENTS

Reviewer #1 (Remarks to the Author):

Kershaw and colleagues determined the cryo-EM structure of soluble mTpoR/Tpo complex at 3.5Å resolution. Consistent with the recent structure of full-length human TpoR/Tpo complex, one Tpo binds to two mTpoR, thereby facilitating receptor dimerization. Specifically, Tpo binds to two distinct binding sites on TpoR, named site 1 and site 2. Site 1 has a high binding affinity and is crucial for TpoR activation. Site 2 has a relatively weak binding affinity, but it facilitates the optimal receptor function. Biochemical and cell biological analysis partly supports the structural observation. Moreover, the authors modeled the binding sites of Romiplostim, a peptide agonist of TpoR, showing that the peptide binding site is greatly overlapped with the Tpo binding site.

The study of the Tpo-dependent activation mechanism of TpoR is important and interesting, as it can provide insight into the development of effective TpoR agonists for therapeutic purposes. However, the current manuscript and Figures are difficult to follow, especially for broad readers of Nature Communications. Biochemical and cellular experiments are not quite as robust.

Specific comments are listed below:

(1) Line 97, as the authors noted, a recent study demonstrated a Tpo-dependent TpoR activation mechanism. This paragraph should be revised.

(2) To generate TpoR/Tpo complex in solution, the authors used TopN only. The binding assay did not show a great difference between Tpo and TopN, but it is true that full length, heavily glycosylated Tpo has a slightly higher affinity for binding to TpoR. Have you tried generating TpoR/Tpo complex with full-length Tpo?

(3) It is difficult to follow the description in the manuscript with a representative image in Figure 4B because most of the residues are not noted. For example, line 282, "F45 is a key part of this hydrophobic network on the receptor", however, it was not noted until Figure 5.

I recommend showing one figure for the overall binding mode as Figure 5 top and Figure 4A, one for site 1 as Figure 4B and Figure 5's site 1 figure, and another for site 2 as Figure 4C,D and Figure 5's site 2 figure.

(4) It will be beneficial to note the salt bridge between R102 of TpoR and D66 of Tpo in the site1, similar to other hydrogen bonds.

(5) In line 341, "As described in Figure 1B, the affinity of Tpo for TpoR was 100 nM KD as determined using surface plasmon resonance."—Did you use Tpo full length or TpoN?

(6) The authors designed Tpo F162E (site 1 mutant) and D29E/Y (site 2 mutant). Please indicate Tpo F162 and D29 in the main figures along with their respective residues of TpoR. It is important to present the quality of the recombinant proteins. It would be beneficial to demonstrate the effect of mutant TpoRs that disrupt site 1 or site 2 interactions specifically.

(7) There is a lack of consistency in the way mutants are generated. For example, they designed F104S and F105A. Why did the authors use F104S instead of F104A? Did you test F104A but did not observe any effect? Is there another reason? Furthermore, it is unclear why they designed D29E since both are negatively charged.

(8) It is difficult to interpret Figure 8E because no residue information is provided in the Alpha fold prediction of TpoR/Romiplostim.

(9) In Figure 9, please note D1, D2, Tpo or Epo.

(10) A description of "site 3" is provided in the discussion, but it should be moved to the result part, and the D4-D4 interaction should be presented as a separate figure (at least in the supplementary figure). As TPO is not involved in this interaction, I am unsure whether they can use the term of site 3.

(11) In discussion, line 511, "the unusual 34 residue insertion within D2..." D2 insertion in Figure 4D is unclear. The figure should be revised and properly cited. It is not clear how the

current structure can provide more accurate predictions regarding loop deletion. It is necessary to revise or eliminate this part.

Minor points

(1) It should be consistent throughout the entire manuscript, whether it is site 1 and site 2 or site I and site II.

(2) The citations to the figures were incorrect, for example, lines 140, 146, 149, and 153.

(3) Line 420, does the monomeric form of the peptide indicate half Romiplostim?

Reviewer #2 (Remarks to the Author):

Sarson-Lawrence and colleagues provide the Cryo-EM structure of the murine Tpo-TpoR complex and perform structural analysis completed by functional assays to validate the acquired structural information. The acquisition of this structure is of great relevance to the field and provides important information that will have a lasting impact. As mentioned by the authors, the resolution of the structure of the murine Tpo-TpoR complex is to be put in perspectives with the very recent publication of the human Tpo-TpoR complex by Tsutsumi and colleagues which occurred just prior to the submission of this manuscript. The structures of human and murine Tpo-TpoR complexes share high homology as expected. However, some differences are noticed, that the authors should compare in more details. In addition, there are many instances in the text where key elements are missing or are inexact. This should be addressed prior to publication.

I would support publication of this manuscript after the following corrections have been made.

1) Comparison with published Cryo-EM structure of human Tpo-TpoR complex: the PDB of this complex is currently available and comparison should be made with it. It should help the reader understand the similarities and differences between the two species, especially

and murine and human MPL/TpoR behave very differently functionally.

2) Unlike in this manuscript where the complex was produced in Expi293F that have a functional glycosylation machinery, the human TpoR-Tpo complex was produced in GnTI- cells that cannot produce mature N-glycans. Given that N-glycosylation of TpoR is highly relevant for activation by mutants of CALR in pathology, and that cell-surface TpoR is mature for N-glycosylation, the authors should assess whether in their system they obtain mature or immature N-glycans? This can be easily done by mass spectrometry. This can also be a difference between this study and the one on human TpoR which was performed with immature TpoR.

3) The authors state the TpoR could form pre-formed dimers by citing Waters et al., 2015 and then that it largely exists as a monomer at the cell surface with some transient homodimerization citing Wilmes et al., Science 2020. This appears contradictory. In their manuscript, Wilmes and colleagues used human TpoR and indeed showed that it is largely present as a monomer with some transient homodimerization. Leroy et al. JBC 2016 (PMID: 26627830) reported that human TpoR is a monomer but that a larger fraction of murine TpoR exists in preformed inactive dimers. This should be discussed.

4) The insertion +2 residues data are interesting and are reminiscent from orientation dependent signaling that has been reported for murine TpoR (Staerk et al., EMBO 2011 PMID: 21892137). Of interest murine TpoR can induce activation in 6 out of 7 dimeric conformations (Staerk et al., EMBO 2011, PMID: 21892137) while human TpoR is active in only 3 out of 7 dimeric conformations (Papadopoulos et al., Blood 2023 PMID: 37616564). This could be discussed related to data obtained on the variant of leucine zipper used to dimerized TpoR (Supplementary Figure 8B).

5) Furthermore, In supplementary Figure 8B, the authors introduced insertions in the leucine zipper and measured binding of Tpo to dimerized TpoR in different conformation. Interestingly, they should that modulation the orientations of this artificial TM domain modified the affinity of Tpo for the pre-formed dimer. First, the authors should add on the same graph the affinity of non-mutated TpoR-Leuz as a comparison. Then, the exact nature of these mutants should be specified (+1A, +2A,...). Finally, it would be highly relevant if the

authors could identify in which conformation does the LeuZ dimerizes naturally (given that it is this orientation that allows maximal TpoR binding) and assess whether this correlates with active orientations previously identified in Staerk et al., 2011. Since the structure of GCN4 is solved, it should be quite straightforward.

6) Ref 31 is a review, the authors should cite original papers instead.

7) Line 140: Figure 1 C,E (numbering is missing).

8) The authors identified mannosylation as a post-translational modification. This is very interesting, especially as it was shown before to regulate TpoR signaling (Sasazawa et al., 2015). This should be discussed.

-

- Minor

- The basis for dimerization of TpoR via TM and juxtamembrane domains in the context of mutants W515L and S505 (presented at page 4) has been established in Defour et al. 2013 PMID: 23359689, Matthews et al. 2013 PMID: 2140271 and 2023 PMID: 37338955.

Response to reviewers

We would like to thank the reviewers for their careful reading of this manuscript and their suggestions. We appreciate their time and effort during the review process. We have responded to the comments in green in the text below, and all major changes to the text are highlighted in red in the revised version of the manuscript.

REVIEWER COMMENTS

Reviewer #1 (Remarks to the Author):

Kershaw and colleagues determined the cryo-EM structure of soluble mTpoR/Tpo complex at 3.5Å resolution. Consistent with the recent structure of full-length human TpoR/Tpo complex, one Tpo binds to two mTpoR, thereby facilitating receptor dimerization. Specifically, Tpo binds to two distinct binding sites on TpoR, named site 1 and site 2. Site 1 has a high binding affinity and is crucial for TpoR activation. Site 2 has a relatively weak binding affinity, but it facilitates the optimal receptor function. Biochemical and cell biological analysis partly supports the structural observation. Moreover, the authors modeled the binding sites of Romiplostim, a peptide agonist of TpoR, showing that the peptide binding site is greatly overlapped with the Tpo binding site.

The study of the Tpo-dependent activation mechanism of TpoR is important and interesting, as it can provide insight into the development of effective TpoR agonists for therapeutic purposes. However, the current manuscript and Figures are difficult to follow, especially for broad readers of Nature Communications. Biochemical and cellular experiments are not quite as robust.

Thank you for these comments – we have changed the text and figures significantly to make it easier to follow. We feel that this has improved readability. The changes are detailed below.

Specific comments are listed below:

(1) Line 97, as the authors noted, a recent study demonstrated a Tpo-dependent TpoR activation mechanism. This paragraph should be revised.

We have revised this paragraph to reference the recent structure by Tsutsumi et al, 2023, Cell. The new line reads as follows:

“Despite 30 years since its discovery, Tpo signalling remained until recently one of the few cytokine/receptor systems that lacked structural characterisation. The structure of the human Tpo:TpoR complex was solved earlier this year at 3.4 Å resolution. Here we present....”

We have also included a detailed comparison of the structures in the Discussion, as detailed later.

(2) To generate TpoR/Tpo complex in solution, the authors used TpoN only. The binding assay did not show a great difference between Tpo and TpoN, but it is true that full length, heavily glycosylated Tpo has a slightly higher affinity for binding to TpoR. Have you tried generating TpoR/Tpo complex with full-length Tpo?

Yes-it is indeed true that there is a slight increase in affinity when using the full-length cytokine. Importantly, this was true in experiments that are insensitive to protein concentration (which is the major source of error in affinity measurements). For example, when the cytokine is bound to the SPR chip and the receptor is passed over the top, Figure 1B. However, it was difficult for us to imagine a biological context when such a small difference would be significant and therefore we did not want to claim that this is biologically meaningful. Given the concern of the reviewer we have modified the text as such:

Original:

*"Tpo and Tpo_N were similarly potent in their affinity for Mpl, with a calculated K_D of 100 ± 20 nM and 160 ± 30 nM respectively (**Figure 1B**). This data implies that the C-terminal domain of Tpo has a negligible contribution to receptor binding."*

Modified:

*"Tpo and Tpo_N were similarly potent in their affinity for Mpl, although the full-length cytokine bound with slightly higher affinity, K_D of 100 ± 20 nM and 160 ± 30 nM respectively (**Figure 1B**). This data implies that the majority of the affinity is encoded by the 4-helix bundle domain of the cytokine."*

In terms of trying to generate a Tpo/TpoR complex in solution, yes we certainly attempted to do this. However, just as for the TpoN version, we were never able to isolate a 2:1 TpoR/Tpo complex unless the receptor was leucine-zippered. The reason for choosing TpoN as the cytokine for our CryoEM studies was the unstructured/heavily glycosylated nature of the Tpo C-terminal domain and the fact that the yield of TpoN was superior. We have added an extra sentence to clarify this.

*"Although reproducible and robust formation of a 1:1 TpoR:Tpo complex at various concentrations and ratios of protein was observed, we were never able to generate the 2:1 stoichiometry (TpoR:Tpo) expected for homodimeric receptors (**Figure 1C,E**). This was true regardless of whether Tpo or TpoN was used."*

(3) It is difficult to follow the description in the manuscript with a representative image in Figure 4B because most of the residues are not noted. For example, line 282, "F45 is a key part of this hydrophobic network on the receptor", however, it was not noted until Figure 5. I recommend showing one figure for the overall binding mode as Figure 5 top and Figure 4A, one for site 1 as Figure 4B and Figure 5's site 1 figure, and another for site 2 as Figure 4C,D and Figure 5's site 2 figure.

Thank you, we have modified these figures as suggested by the reviewer and agree that it is much improved.

(4) It will be beneficial to note the salt bridge between R102 of TpoR and D66 of Tpo in the site1, similar to other hydrogen bonds.

Thank you, we have modified the figure as suggested by the reviewer.

(5) In line 341, "As described in Figure 1B, the affinity of Tpo for TpoR was 100 nM K_D as determined using surface plasmon resonance."—Did you use Tpo full length or TpoN?

Sentence changed from:

"As described in Figure 1B, the affinity of Tpo for TpoR was 100 nM K_D as determined using surface plasmon resonance."

To:

"As described in Figure 1B, the affinity of Tpo and Tpo(N) for TpoR was 100nM and 160nM respectively as determined using surface plasmon resonance."

(6) The authors designed Tpo F162E (site 1 mutant) and D29E/Y (site 2 mutant). Please indicate Tpo F162 and D29 in the main figures along with their respective residues of TpoR.

Thank you, this was an oversight of ours and we have now labelled these residues in Figures 4 and 5 and have also added a more detailed view in an additional figure, supplementary figure 11.

It is important to present the quality of the recombinant proteins. It would be beneficial to demonstrate the effect of mutant TpoRs that disrupt site 1 or site 2 interactions specifically.

Thank you, we agree and have added a supplementary figure (supplementary figure 11) showing an SDS-PAGE gel of all the recombinant mutants.

We also agree that it would be interesting to produce mutants of TpoR that affect specifically Site I and Site II, however the binding site on the two receptor chains are essentially identical and we were unable to design a mutant that would only affect one of them.

(7) There is a lack of consistency in the way mutants are generated. For example, they designed F104S and F105A. Why did the authors use F104S instead of F104A? Did you test F104A but did not observe any effect? Is there another reason? Furthermore, it is unclear why they designed D29E since both are negatively charged.

Although mutation to alanine is generally accepted as a standard choice for functional mutagenesis, we specifically chose F104S rather than F104A because F104S is a reported clinical mutation. For F105, we used the standard conversion to alanine.

The sidechain of D29 is adjacent to a hydrophobic pocket formed by TpoR V255, L257 and F164. Therefore we chose residues whose extra-length (Glu and Tyr) would clash with those residues. The choice of Glutamate was so that charge could be maintained as D29 orients Tpo R31 which hydrogen bonds to TpoR D53. We appreciate that we did not explain our logic well in the manuscript and have now added this detail into Supplementary Figure 12.

(8) It is difficult to interpret Figure 8E because no residue information is provided in the Alpha fold prediction of TpoR/Romiplostim.

Thank you, we have modified the figure to add this information.

(9) In Figure 9, please note D1, D2, Tpo or Epo.

Thank you, we have modified the figure to add these labels and agree it is much more informative.

(10) A description of "site 3" is provided in the discussion, but it should be moved to the result part, and the D4-D4 interaction should be presented as a separate figure (at least in the supplementary figure). As TPO is not involved in this interaction, I am unsure whether they can use the term of site 3.

We appreciate the reviewers remark regarding the term "site III" and have removed it throughout the manuscript, instead referring to it as the D4 interface. This interaction is now described in the results section as well as being mentioned in the discussion. In addition, we have added a new supplementary figure showing D4:D4 interaction (Supplementary Figure 6).

Results:

"Despite the poor resolution, it appears that D4 contributes to homodimerization through a D4:D4 interface with an estimated buried surface area of approximately 110Å² per chain (PISA [50]) (Supplementary Figure 6)."

Discussion:

"Instead, an interface is formed through D4-D4 interactions in the membrane proximal CHR. This D4-D4 interaction in the Tpo/TpoR complex (~110 Å² buried surface) is not as extensive as seen for PRL and GHR....."

(11) In discussion, line 511, "the unusual 34 residue insertion within D2..." D2 insertion in Figure 4D is unclear. The figure should be revised and properly cited. It is not clear how the current structure can provide more accurate predictions regarding loop deletion. It is necessary to revise or eliminate this part.

We have now referenced Figure 3 for that statement, and we have also added in a new Supplementary Figure (Figure S14) that highlights the D2 insertion.

The difficulty in modelling deletions that could safely remove the unstructured insertion arises from the fact that the disulphide bond partner of C211 in the insertion was unknown (it is a cysteine in D3). Thus, deletion of the entire loop would remove the linkage to D3 as well as the contact with cytokine whereas now we can design a deletion that only removes one or the other. However, given this will be of limited interest to the readers of Nature Communications we have deleted that sentence.

(1) It should be consistent throughout the entire manuscript, whether it is site 1 and site 2 or site I and site II.

We have corrected this to site I and site II throughout the manuscript.

(2) The citations to the figures were incorrect, for example, lines 140, 146, 149, and 153. We apologise for these errors and have corrected these references to the figures.

(3) Line 420, does the monomeric form of the peptide indicate half Romiplostim? Yes, it does, and we have added the peptide sequences in the main text to clarify.

Reviewer #2 (Remarks to the Author):

Sarson-Lawrence and colleagues provide the Cryo-EM structure of the murine Tpo-TpoR complex and perform structural analysis completed by functional assays to validate the acquired structural information. The acquisition of this structure is of great relevance to the field and provides important information that will have a lasting impact. As mentioned by the authors, the resolution of the structure of the murine Tpo-TpoR complex is to be put in perspectives with the very recent publication of the human Tpo-TpoR complex by Tsutsumi and colleagues which occurred just prior to the submission of this manuscript. The structures of human and murine Tpo-TpoR complexes share high homology as expected. However, some differences are noticed, that the authors should compare in more details. In addition, there are many instances in the text where key elements are missing or are inexact. This should be addressed prior to publication.

I would support publication of this manuscript after the following corrections have been made.

1) Comparison with published Cryo-EM structure of human Tpo-TpoR complex: the PDB of this complex is currently available and comparison should be made with it. It should help the reader understand the similarities and differences between the two species, especially and murine and human MPL/TpoR behave very differently functionally.

The PDB coordinates of the human complex have now been released and we were able to compare the structures in detail. We have added the following section (page 29).

"Comparison of our murine TPO:TPOR structure with that of the human TPO:TPOR complex [72], both of which were obtained at comparable resolution, reveals that they are remarkably similar (RMDS 2.0 for C-alpha over the entire complex Supplementary Figure 14), despite differences in glycosylation and quite different approaches to obtaining the structure. In this study we used expi293 cells to make TpoR and showed that the glycans were mature glycoforms. Tsutsumi et al. [72] used 293GnT- cells, expected to produce immature, high-mannose glycans. While our approach used a Leucine-zippered construct to generate dimeric recombinant ectodomain, Tsutsumi et al. [72] used the full-length receptor extracted into detergent micelles. Neither the leucine zipper nor the transmembrane domain were visible in either structure. Both studies used only the cytokine bundle of Tpo, with the majority of the C-terminal unstructured region removed, however Tsutsumi et al. used a cross-linking agent to stabilise the cytokine:receptor complex, whereas in our study we were able to isolate the complex without the need for cross-linking.

The mouse and human receptors share 78% sequence identity (83% similarity) within their ectodomain. These sequence differences are located uniformly throughout the receptor and four of them are within 4.5 angstroms of site I and/or site II. Of these, mTpoR R70 (Proline in hTpoR) forms a hydrogen bond with Tpo Q126 and D99 (Glu in hTpoR) hydrogen bonds with Tpo R99. Equivalent interactions are not present in the human Tpo/TpoR structure. The biggest difference between the two species is the D2 insertion. The human insertion is eight residues

longer and shares only 42% homology with the mouse sequence. However, this insertion was not resolved in the human structure and was indeed only weakly resolved in in one chain of the structure presented here, where it was seen to contact cytokine, forming site IIb. We observe that, structurally, the insertion contacts cytokine at one end and domain 3 (to which it is disulphide bonded) at the other. The latter disulphide was not resolved in the hTpo/TpoR structure due to poorer resolution of the insertion. Interestingly, there is only a single cysteine not engaged in a disulphide bond, C413, in domain 4. This cysteine, which is conserved in placental mammals but not in other species, is located 13 angstroms from its counterpart in the other receptor chain. Even though flexibility is seen in the legs of the receptor that can bring the two D4 domains slightly closer together (Supplementary Movie 1), it seems unlikely that they would disulphide bond in vivo."

Further differences are noted in response to comments (3) and (4) below

2) Unlike in this manuscript where the complex was produced in Expi293F that have a functional glycosylation machinery, the human TpoR-Tpo complex was produced in GnTI- cells that cannot produce mature N-glycans. Given that N-glycosylation of TpoR is highly relevant for activation by mutants of CALR in pathology, and that cell-surface TpoR is mature for N-glycosylation, the authors should assess whether in their system they obtain mature or immature N-glycans? This can be easily done by mass spectrometry. This can also be a difference between this study and the one on human TpoR which was performed with immature TpoR.

Thank you for this suggestion. We have performed a mass spectroscopy analysis of the glycosylation and have shown that, as expected for the expi293 expression system, we have mature glycans. We have noted this in both the results and the discussion:

Results:

There was clear density for three N-linked glycosylation modifications in both chains at N117, N178 and N349, and additional density at Cδ of W465 consistent with mannosylation at this site (Supplementary Figure 7). Glycosylation state was validated by mass spectrometry (Supplementary Table 2), which confirmed there are three modified asparagine residues, with low oligomannose content, indicating that the glycans are in the mature glycoforms expected from expi293 cells. Tryptophan mannosylation was detected on several tryptophans, including W465 and W468 in the WSAWS motif (Supplementary Figure 8), consistent with observations of the human protein [42, 51].

Discussion:

In this study we used expi293 cells to make TpoR and showed that the glycans were mature glycoforms. Tsutsumi et al. [72] used 293GnTI- cells, expected to produce immature, high-mannose glycans[75].

3) The authors state the TpoR could form pre-formed dimers by citing Waters et al., 2015 and then that it largely exists as a monomer at the cell surface with some transient homodimerization citing Wilmes et al., Science 2020. This appears contradictory. In their manuscript, Wilmes and colleagues used human TpoR and indeed showed that it is largely present as a monomer with some transient homodimerization. Leroy et al. JBC 2016 (PMID: 26627830) reported that human TpoR is a monomer but that a larger fraction of murine TpoR exists in preformed inactive dimers. This should be discussed.

We have now clarified the sentence in the introduction that is referred to here, and added a section in the discussion, citing Leroy et al JBC 2015:

“Formation of the active dimeric TpoR receptor appears to depend on multiple contact points: receptors are bridged by cytokine, and contact each other at D4, the transmembrane helix and JAK pseudokinase domain [61, 75]. The propensity of the receptor to dimerise in the absence of cytokine is higher for mouse than for human, and it has been shown that removal of the ectodomain of mTpoR (but not hTpoR) results in a further increase in receptor dimerization [76]. Given the similarity in the binding sites and overall conformation of the mouse and human Tpo:TpoR ectodomain complexes (Supplementary Figure 14), it seems likely that the propensity to dimerise or not is a result of the properties of the transmembrane region itself, rather than differences in the ectodomain. In particular His 499, which is Leu in mTpoR, has been shown to limit the formation of pre-formed dimers in the human protein by altering the structure of the transmembrane domain and breaking the helical nature [76], which is presumably restored in the active conformation when cytokine induces dimerisation. Obtaining a structure of the full-length receptor in the presence of cytokine and JAK presents the next structural biology challenge in understanding the details of this system.”

4) The insertion +2 residues data are interesting and are reminiscent from orientation dependent signaling that has been reported for murine TpoR (Staerk et al., EMBO 2011 PMID: 21892137). Of interest murine TpoR can induce activation in 6 out of 7 dimeric conformations (Staerk et al., EMBO 2011, PMID: 21892137) while human TpoR is active in only 3 out of 7 dimeric conformations (Papadopoulos et al., Blood 2023 PMID: 37616564). This could be discussed related to data obtained on the variant of leucine zipper used to dimerized TpoR (Supplementary Figure 8B).

We also noted the similarity between the data from our insertion mutants and the data shown in the above references. Indeed the reason we made those mutations was because of that data. However as our structure was solved it became apparent that there is flexibility in vitro between the TM domain and the ectodomain. This is seen in our structure (which replaced the TM domain with a leucine zipper) and in the human TpoR structure (which included the TM domain) because in both cases no density was observed past Ser475 (eight residues upstream of the tryptophan that marks the beginning of the TM domain). Therefore, in the absence of membrane, it is unlikely that difference in cytokine binding we see with the various insertion mutants in vitro is due to a mechanism similar to that seen in cells when the TM domain helix is extended inside the cell. The one insertion mutant that had an effect is the one that places

the ectodomains furthest apart (new Supplementary Figure 10c) so it appears more likely that this slightly hinders the two receptor chains coming together in the correct geometry rather than being a helical register issue per se. We have summarised this in the following passage, and would like to thank the reviewer for reminding us to cite those studies-this was an oversight on our part as we are very familiar with those excellent papers.

Previous studies have shown that certain conformations of the TpoR do not support full signalling throughput [70] and that human TpoR is more restricted than murine TpoR in this regard [71] due largely to the presence of His499 in the human protein. We constructed a limited series of insertion mutants in the leucine zipper to determine whether changes in the helical register of the junction between the ectodomain and the leucine-zipper would affect binding of the cytokine. Although we observed impaired binding by one such mutant, it is unclear whether that is due to a specific conformation or is merely due to that particular construct placing the ectodomains furthest apart (Supplementary Figure 10). There is enough flexibility, in vitro, between the ectodomain and the leucine zipper (or the transmembrane domain [72]) such that no density for these regions, or several residues that precede it, were visible in either TpoR structure [72]. Therefore, any conformational change transmitted from the ectodomain to the transmembrane domain upon cytokine binding seems unlikely in solution. In vivo, it is possible that there is a subtle re-orientation of the receptor on the cell-surface or the membrane itself that could play a role in signal transduction, for example by driving the D4:D4 interaction. Indeed, there is a concentration of arginines on the membrane proximal end of D4 and a tryptophan at the membrane junction which could aid this. Such an interaction occurs immediately downstream of the transmembrane domain, in the juxtamembrane sequence and is important for preventing aberrant activation in the absence of cytokine ([36, 73, 74]). Another possible explanation of the low EC50 in vivo is that the maximum biological response by Tpo is generated by only fractional receptor occupancy. Indeed, a recent study observed a 1 nM EC50, similar to the in vitro KD, when STAT3 phosphorylation (rather than biological outcome) was used as a read-out [61].

5) Furthermore, In supplementary Figure 8B, the authors introduced insertions in the leucine zipper and measured binding of Tpo to dimerized TpoR in different conformation. Interestingly, they should that modulation the orientations of this artificial TM domain modified the affinity of Tpo for the pre-formed dimer. First, the authors should add on the same graph the affinity of non-mutated TpoR-Leuz as a comparison.

Then, the exact nature of these mutants should be specified (+1A, +2A,...).

Thank you for the suggestion, we have made these changes (Supplementary Figure 10).

Finally, it would be highly relevant if the authors could identify in which conformation does the LeuZ dimerizes naturally (given that it is this orientation that allows maximal TpoR binding) and assess whether this correlates with active orientations previously identified in Staerk et al., 2011. Since the structure of GCN4 is solved, it should be quite straightforward.

Our response to this point is merged with our response to point 4, above and we have illustrated the predicted orientation in a new supplementary figure (Supplementary Figure 10c).

6) Ref 31 is a review, the authors should cite original papers instead.

We have replaced this review with three original research papers:

31. Nangalia, J., et al., *Somatic Mutations in Myeloproliferative Neoplasms with Nonmutated JAK2*. *New England Journal of Medicine*, 2013. **369**(25): p. 2391-2405.
32. Klampfl, T., et al., *Somatic Mutations of Calreticulin in Myeloproliferative Neoplasms*. *New England Journal of Medicine*, 2013. **369**(25): p. 2379-2390.
33. Saito, Y., et al., *Calreticulin functions as a molecular chaperone for both glycosylated and nonglycosylated proteins*. *Embo Journal*, 1999. **18**(23): p. 6718-6729.

7) Line 140: Figure 1 C,E (numbering is missing).

Apologies, this is now corrected.

8) The authors identified mannosylation as a post-translational modification. This is very interesting, especially as it was shown before to regulate TpoR signaling (Sasazawa et al., 2015). This should be discussed.

When we performed the glycan finger-printing, we re-examined the mannosylation status. We found evidence of additional mannosylation sites (Supplemental Table 2) and have updated the text and included a reference to Sasazawa et al., 2015. However, Sasazawa et al., examined the effect of mannosylation indirectly by showing that substitution of the mannosylated tryptophan for phenylalanine impacted cell surface expression of TpoR. The four sites that were shown to be essential for signalling are also part of a W-R-W-R-W stack that is a conserved structural motif in type I cytokine receptors (REF). It is possible that the W to F mutation destabilises the protein, rather than a strict requirement for mannosylation itself impacting signalling or surface expression. We have therefore chosen not to comment further on the importance of the mannosylation we observed, beyond noting its presence in mouse TpoR, consistent with previous observations of the human protein. See response to comment (2) for details of changes to text.

•Minor

•The basis for dimerization of TpoR via TM and juxtamembrane domains in the context of mutants W515L and S505 (presented at page 4) has been established in Defour et al. 2013 PMID: 23359689, Matthews et al. 729327 and 2023 PMID: 37338955.

Thank you, we have now cited additional works, as suggested.

REVIEWERS' COMMENTS

Reviewer #1 (Remarks to the Author):

In this revised version of their manuscript, the authors have significantly re-organized the manuscript providing better emphasis on the points retained from the previous manuscript. Additionally, the authors discussed the similarities and differences between the human and mouse TpoR/Tpo complexes. These changes greatly enhance the manuscript. The authors have satisfactorily addressed my concerns.

Minor points.

The manuscript is too long and contains a number of errors in the text, figures, and figure legends. The following are a few examples, but they are not limited.

(1) It should be consistent throughout the entire manuscript, whether TpoR:Tpo complex (signaling) or TpoR/Tpo complex (signaling).

(2) In Figure 2C, transmembrane domains and intracellular domains are not resolved. Note that transmembrane domains are unstructured or remove the cartoon.

(3) In Figure 3A, it is better to use the same color code as in Figure 2B.

(4) Figure legends 3C and 3E should be checked. It does not correspond to the figures.

(5) Figure legends should be revised extensively. In Figure 6, it is unclear what "The top concentration of analyte used in each case was 250 nM, 16 nM, 15 nM, 250 nM, respectively, preceded by four 2-fold dilutions." means.

Reviewer #2 (Remarks to the Author):

I think the manuscript can be accepted for publication. It has been re-written and revised so that missing points are now added and the flow is clear.

Below, we have detailed our responses to the minor points raised by reviewer 1. Reviewer 2 did not request any changes.

Reviewer 1:
Minor points.

The manuscript is too long and contains a number of errors in the text, figures, and figure legends. The following are a few examples, but they are not limited.

We recognise that the manuscript is long and have shortened the manuscript slightly, by removing text from both the introduction and discussion. However, the lengthy nature of the discussion was in part due to incorporation of responses to relevant comments from reviewer 2 and thus we are reluctant to cut this back further unless specifically requested by the editor.

We hope we have now found and corrected any errors in the manuscript.

(1) It should be consistent throughout the entire manuscript, whether TpoR:Tpo complex (signaling) or TpoR/Tpo complex (signaling).

We thank the reviewer for noticing this inconsistency and have changed this to Tpo:TpoR throughout the manuscript.

(2) In Figure 2C, transmembrane domains and intracellular domains are not resolved. Note that transmembrane domains are unstructured or remove the cartoon.

We have noted that the transmembrane domains are unstructured and made this region slightly transparent to remove emphasis. However, we think it is beneficial to keep this part of the cartoon in order to illustrate how it would assemble with respect to the membrane.

(3) In Figure 3A, it is better to use the same color code as in Figure 2B.

We have changed the colour to match one of the TpoR chains in Fig2B (and elsewhere).

(4) Figure legends 3C and 3E should be checked. It does not correspond to the figures.

We have corrected this error.

(5) Figure legends should be revised extensively. In Figure 6, it is unclear what "The top concentration of analyte used in each case was 250 nM, 16 nM, 15 nM, 250 nM, respectively, preceded by four 2-fold dilutions." means.

We have revised the figure legends, paying particular attention to description of SPR experiments.